# Glycine-serine-rich effector PstGSRE4 in *Puccinia striiformis* f. sp. *tritici* inhibits the activity of copper zinc superoxide dismutase to modulate immunity in wheat

**Cong Liu, Yunqian Wang, Yanfeng Wang, Yuanyuan Du, Chao Song, Ping Song, Qian Yang, Fuxin He, Xingxuan Bai, Lili Huang, Jia Guo\*, Zhensheng Kang\*, Jun Guo ⬦ \***

State Key Laboratory of Crop Stress Biology for Arid Areas, College of Plant Protection, Northwest A&F University, Yangling, Shaanxi, P. R. China

\* guojia1889@nwafu.edu.cn (JG); kangzs@nwsuaf.edu.cn (ZK); guojunwgq@nwsuaf.edu.cn (JG)

## Abstract

*Puccinia striiformis* f. sp. *tritici* (*Pst*) secretes an array of specific effector proteins to manipulate host immunity and promote pathogen colonization. In a previous study, we functionally characterized a glycine-serine-rich effector PstGSRE1 with a glycine-serine-rich motif (m9). However, the mechanisms of glycine-serine-rich effectors (GSREs) remain obscure. Here we report a new glycine-serine-rich effector, PstGSRE4, which has no m9-like motif but inhibits the enzyme activity of wheat copper zinc superoxide dismutase TaCZSOD2, which acts as a positive regulator of wheat resistance to *Pst*. By inhibiting the enzyme activity of TaCZSOD2, PstGSRE4 reduces $H_2O_2$ accumulation and HR areas to facilitate *Pst* infection. These findings provide new insights into the molecular mechanisms of GSREs of rust fungi in regulating plant immunity.

**Data Availability Statement:** All relevant data are within the manuscript and its Supporting Information files.

## Author summary

*Pst* secretes numerous effectors to modulate host defense systems. However, the mechanisms of these effectors, especially for glycine-rich or serine-rich effectors, remain obscure. In this study, we identified a new glycine-serine-rich effector, PstGSRE4, which exhibits unusual biochemical properties and is highly induced during early stages of infection. Transgenic expression of *PstGSRE4*-RNAi constructs in wheat significantly reduced virulence of *Pst* and increased $H_2O_2$ accumulation in wheat. Overexpression of *PstGSRE4* in wheat significantly increased virulence of *Pst* and reduced $H_2O_2$ accumulation in wheat. PstGSRE4 was shown to target the ROS-associated regulatory factor TaCZSOD2, which was proved as a positive regulator of wheat immunity in this study. Further study revealed that PstGSRE4 inhibited the enzyme activity of TaCZSOD2 and thus compromises the host immune systems. This work reveals a novel strategy that rust fungi exploit to modulate host defense and facilitate pathogen infection.

**Funding:** This study was financially supported by the National Key Research and Development Program of China to JG (2021YFD1401000) (http://www.most.gov.cn/), the National Natural Science Foundation of China to JG (32172381 and 31972224) (https://www.nsfc.gov.cn/), Key Research and Development Program of Shaanxi to JG (2021ZDLNY01-01) (https://kjt.shaanxi.gov.cn/), Natural Science Basic Research Program of Shaanxi to JG (2020JZ-13) (https://kjt.shaanxi.gov.cn/) and the 111 Project from the Ministry of Education of China to ZK (B07049) (http://www.moe.gov.cn/). The funders had no role in study design, data collection and analysis, decision to publish, or preparation of the manuscript.

**Competing interests:** The authors have declared that no competing interests exist.

## Introduction

In nature, plants are exposed to a variety of biotic and abiotic stresses, including the invasion of numerous pathogenic microorganisms. In their interactions, plants and pathogens confront processes of defense and pathogenicity and co-evolve. Upon pathogen infection, pattern recognition receptors (PRRs) in plants recognize the pathogen-associated molecular pattern (PAMP) and activate PAMP-triggered immunity (PTI) to form the first level of defense [1]. Pathogens have formed a large number of virulence factors during the long-term evolution with the host, and successfully infect and colonize the host by acting on the host plant cells [1]. Effectors, as a type of very important virulence factors, are secreted from the pathogen into the host primarily to inhibit the host's defense response, and thus cause host plant susceptibility. In addition, when certain avirulence effectors from the pathogen are directly or indirectly recognized by plant disease-resistant proteins, the plant immune system is strongly activated to induce the host cell hypersensitive response (HR), which has been termed effector-triggered immunity (ETI) [2]. Therefore, the effectors of a pathogen have a dual function of virulence and avirulence, which is not only an important weapon of pathogenicity, but also an important target of the plant immune system. Both PTI and ETI include the induction of reactive oxygen species (ROS), a key component of the defense system [3,4].

The sharp increase of ROS is a common manifestation when plants are confronted with various pathogens, indicating that ROS play a vital role in the process of plant resistance to pathogens. ROS burst is generally defined as a rapid production of high levels of ROS in response to external stimuli [5]. Superoxide radicals ($O_2^-$) and hydrogen peroxide ($H_2O_2$) are considered important ROS in response to biotic stress [6]. During plant-pathogen interaction, penetration of pathogen into host plasma membrane triggers the early $O_2^-$ burst by an NADPH oxidase, then they are rapidly converted to $H_2O_2$ by dismutation [5]. Most of the data seem to indicate that the major ROS building the oxidative burst is $H_2O_2$, with possible participation of $O_2^-$. On the one hand, $O_2^-$ and $H_2O_2$ are directly toxic to pathogens. For instance, the accumulation of $O_2^-$ or $H_2O_2$ caused by *Pseudomonas syringae* pv. *tabaci* significantly decreased the number of bacteria in *Nicotiana benthamiana*, and then the number of bacteria significantly increased following the addition of SOD or other reactive oxygen scavengers [7]. On the other hand, $H_2O_2$ can also act as signaling molecules to directly or indirectly activate the expression of resistance genes and defense genes. $H_2O_2$ can induce the increase of antioxidant enzyme activity in plants to resist the invasion of pathogens. Exogenous $H_2O_2$ can induce a significant increase in glutathione S-transferase (GST) transcription in soybean suspension cells, and $H_2O_2$ scavengers can prevent this effect [8]. In addition, $H_2O_2$ also participate in the lignification of cell walls and the cross-linking of proteins to cell walls to strengthen plant cell walls against pathogen invasion. After infection with diseased substances, synthesis of $H_2O_2$ was observed in lignification sites of plant tissues [7]. $H_2O_2$ can also induce the occurrence of plant HR response. A large number of experiments proved that exogenous $H_2O_2$ can induce HR in cells of *Arabidopsis thaliana* [9]. Interestingly, many studies have shown that effector proteins can control the host immune response by interfering with the host ROS signaling pathway [10–13]. Understanding the mechanism of effectors regulating ROS-related targets will increase our knowledge of molecular mechanisms underlying the interaction between plants and phytopathogens, and provide a theoretical foundation to achieve durable disease resistance.

Superoxide dismutase (SOD) is an important component of the antioxidant enzyme system and is widely distributed in microorganisms, plants and animals. It catalyzes superoxide anion ($O_2^-$) radical disproportionation to produce $O_2$ and $H_2O_2$, and plays an important role in the balance between oxidation and oxidation resistance [14]. Based on their metal cofactors,

protein folds, and subcellular distribution, SODs are mainly categorized as CuZnSODs, FeS-ODs, and MnSODs [15]. A previous study indicated that infection of grape with grapevine fanleaf virus caused the accumulation of ROS and activated its enzyme defense system, including SOD [16]. Among the isoenzymes of SOD in sunflower, the expression of *CuZnSOD* under biological stress is the most affected, indicating that CuZnSOD is the main antioxidant defense enzyme [17]. When the *CuZnSOD* gene in tomato chloroplasts was transferred into two *N. benthamiana* strains, it enhanced the resistance to *anthrax* by changing the expression of the antioxidant enzyme [18]. In the *Phaseolus vulgaris-Uromyces appendiculatus* interaction, the expression of CuZnSOD was increased greatly during the incompatible interaction [19]. Recently, in the study of barley-powdery mildew interaction, loss-of-function mutations in *Mla* and *Rar1* both resulted in the reduced accumulation of copper-zinc superoxide dismutase 1 (*HvSOD1*), whereas loss of function in *Rom1* re-established *HvSOD1* levels [20]. In the study of rice-*Magnaporthe oryzae* interaction, different SODs in *miR398b* regulated resistance to rice blast disease, and *miR398b* increased total SOD activity to upregulate the $H_2O_2$ concentration and thereby improve disease resistance [21]. However, there have been no reports on phytopathogenic effectors targeting and regulating CuZnSODs from plants to suppress host immune response.

Among the diseases caused by rust fungi, the diseases on *Gramineae* and *Leguminosae* seriously threaten the safety of food production in China and throughout the world [22]. Stripe rust is one of the most serious diseases of wheat in the world [23]. Wheat has evolved resistance genes to protect against disease. However, *Pst* constantly mutates to overcome these resistance genes, and the effectors contributed significantly to the virulence diversity of *Pst* [24]. Due to the importance of effector proteins in the interaction between pathogens and plants, more and more attention has been paid to the study of effector proteins. Recently, stripe rust effector Pst18363 has been reported to stabilize a negative regulator of wheat defense, TaNUDX23, which suppresses ROS accumulation and facilitates *Pst* infection [25]. Another stripe rust effector, Pst_12806, is translocated into chloroplasts and perturbs photosynthesis, avoiding triggering cell death and supporting pathogen survival on living plants [26]. In several organisms, glycine- or serine-rich proteins have been shown to participate in RNA splicing, metabolism and signal transduction [27,28]. Pathogen effectors with a high content of glycine or serine could potentially modify the host's metabolism or signal transduction [29]. In *Pst*, a glycine-serine-rich effector protein PstGSRE1 containing a glycine-serine-rich motif (m9) has been shown to disrupt the nuclear localization of TaLOL2 and suppress ROS-mediated cell death induced by TaLOL2, thus compromising host immunity [29]. However, the mechanisms of glycine-serine-rich effectors, remain obscure, and further investigation is required. In this study, we characterized a new glycine-serine-rich effector protein PSTCY32_07414 (alias PstGSRE4), which lacks the m9-like motif, targets a wheat copper zinc superoxide dismutase TaCZSOD2. PstGSRE4 is required for full virulence of *Pst* in wheat. Further analyses showed that TaCZSOD2 is a positive regulator of wheat resistance to *Pst*, and PstGSRE4 reduces $H_2O_2$ accumulation by inhibiting the activity of TaCZSOD2 to facilitate *Pst* infection. Our results provide new insights into the molecular mechanisms of glycine-serine-rich effectors of rust fungi regulating host immunity.

## Results

### PstGSRE4 is a glycine-serine-rich effector protein lacking m9-like motif in *Pst*

In our previous study, four glycine-serine-rich effectors were identified from *Pst* [29]. Sequence analysis showed that PSTCYR32_07414 encodes a 232-amino-acid secreted protein

which was enriched in glycine (12.93%) and serine (19.40%) and did not contain any known functional domains except for a 22 amino acid (aa) signal peptide at its N-terminus (**S1A Fig**). BLASTp analyses revealed that homologs of PstGSRE4 can be found only in rust fungi, including 13 in *Pst*, seven in *Puccinia triticina* (*Pt*) and four in *Puccinia graminis* f. sp. *tritici (Pgt)* (**S1 and S2 Tables**), indicating that GSREs constitute a large family within the rust fungi. Using MEME, we found that PSTCYR32_07414 did not have the motif as the m9 motif of PstGSRE1 (**S1B Fig**). Moreover, yeast two-hybrid (Y2H) assay indicated that PstGSRE4 does not interact with TaLOL2 (**S1C Fig**), suggesting that it is functionally diverged. Thus, PSTCYR32_07414 was designated *Puccinia striiformis* Glycine-Serine-Rich Effector 4 (PstGSRE4) and selected for further study.

## Relative transcript levels of *PstGSRE4* at different *Pst* infection stages

In order to characterize the expression pattern of *PstGSRE4* during *Pst* infection stages, we analyzed its relative transcript levels by qRT-PCR at different time points during the infection process. We used fresh ungerminated urediniospores of CYR32 and infected wheat tissues collected from 6 to 264 hpi to detect the transcripts of *PstGSRE4*. qRT-PCR assays showed that *PstGSRE4* was upregulated during *Pst* infection (6–72 hpi), and reached higher expression levels at 24–48 hpi (approximately 158-fold, 145-fold, and 140-fold, respectively), corresponding to the formation of the haustorium (**S2 Fig**). The transcript levels were downregulated at the late 'parasitic/biotroph' stage (168 hpi) and 'sporulation' stage (216–264 hpi) (**S2 Fig**).

## PstGSRE4 suppresses Pst322- and Bax-induced cell death by decreasing H$_2$O$_2$ accumulation

SignalP 5.0 analysis showed that PstGSRE4 has a signal peptide encoded by the first 22 amino acids. Secretion of PstGSRE4 was verified through a signal sequence trap system [29]. pSUC2T7M13ORI-PstGSRE4 was transferred into the yeast SUC2-minus strain, YTK12. The fusion of the signal peptide of PstGSRE4 to the mature sequence of SUC2 promoted the successful secretion of invertase, which enables the yeast cells to hydrolyze raffinose and grow on YPRAA media (**S3 Fig**). In addition, we found that the TTC-treated PstGSRE4 culture filtrates turned red, confirming invertase activity (**S3 Fig**). The oomycete effector Avr1b was used as positive control, and the YTK12 strains with or without pSUC2 vector were used as negative controls. These results indicate that the signal peptide of PstGSRE4 is functional.

To examine the function of PstGSRE4, we used agro-infiltration to transiently express it in *N. benthamiana*. We observed that PstGSRE4 inhibited an elicitor-like protein [28] Pst322-induced cell death (**S4A Fig**). In addition, PstGSRE4 suppressed the pro-apoptotic protein Bax-induced cell death (**S4B Fig**). Accumulation of PstGSRE4, Pst18363, eGFP, Pst322 and Bax proteins in infiltrated tissue were confirmed by Western blots (**S4C Fig**).

Because H$_2$O$_2$ is a crucial trigger of cell death and PstGSRE4 inhibits Bax- or Pst322-triggered cell death, we also test if PstGSRE4 suppresses Bax- or Pst322-triggered cell death by preventing H$_2$O$_2$ accumulation. DAB staining was used to examine the H$_2$O$_2$ levels in the infiltrated leaves. The DAB staining in the leaf regions infiltrated with Bax or Pst322 accumulated a large number of H$_2$O$_2$ compared with those infiltrated with MgCl$_2$ buffer (**S5A and S5B Fig**). And the DAB staining in the leaf regions infiltrated with PstGSRE4/Bax or PstGSRE4/Pst322 was much weaker compared with those infiltrated with buffer/Bax or buffer/Pst322 (**S5C and S5D Fig**). These results suggested that PstGSRE4 can suppress Bax- or Pst322-triggered cell death by preventing H$_2$O$_2$ accumulation.

## PstGSRE4 suppresses callose deposition and *Pst*-induced $H_2O_2$ accumulation

The important function of fungal effectors is to suppress PTI and/or ETI in order to create a suitable environment for infection. To analyze the ability of PstGSRE4 to inhibit PAMP-triggered responses, we used the bacterial type III secretion assay [29] to deliver *PstGSRE4* into wheat leaves (**S6A Fig**). Callose deposition triggered by *Pseudomonas fluorescence* strain EtHAn carrying pEDV6-*PstGSRE4* was significantly reduced at 24 and 48 hpi compared with the control EtHAn with or without pEDV6-RFP (**S6B and S6C Fig**). We further analyzed $H_2O_2$ accumulation triggered by *Pst* race CYR23 which is incompatible with Suwon11. The results showed that after transient transformation, $H_2O_2$ accumulation was significantly suppressed with EtHAn-*PstGSRE4* at 24 and 48 hpi compared with the control EtHAn with or without pEDV6-RFP (**S6D and S6E Fig**). These results indicated that delivering *PstGSRE4* into wheat inhibits PTI-associated callose deposition as well as $H_2O_2$ accumulation in wheat.

## Silencing of *PstGSRE4* reduces virulence of *Pst*

To test whether *PstGSRE4* is involved in pathogenicity of *Pst*, host-induced gene silencing (HIGS) mediated by barley stripe mosaic virus (BSMV) was used to silence *PstGSRE4* expression in *Pst* during the infection process. Two silencing fragments of *PstGSRE4* were designed to generate two different BSMV constructs (**S7A Fig**) for specifically silencing *PstGSRE4*. All of the wheat leaves infected with BSMV:γ, BSMV:PstGSRE4-1/2as expressed similar phenotypes of mild chlorotic mosaic symptoms at 12 d post-inoculation (dpi), whereas mock-inoculated leaves remained green and healthy (**S7B Fig**). Obvious photo-bleaching was observed in the BSMV:TaPDS-inoculated wheat, indicating that the BSMV-HIGS system functioned effectively (**S7B Fig**). Subsequently, the fourth leaves of BSMV-inoculated wheat were inoculated with freshly collected ungerminated urediniospores of CYR32. The number of rust pustules was significantly reduced in the wheat leaves inoculated with BSMV:PstGSRE4 compared with the leaves of BSMV:γ-inoculated wheat (**S7B Fig**). qRT-PCR analysis of total RNA extracted from silenced leaves, which were sampled at 24, 48 and 120 hpi, revealed that the expression of *PstGSRE4* was significantly reduced in the wheat leaves inoculated with BSMV:PstGSRE4 (**S7C Fig**). The biomass of *Pst*/wheat also decreased in the leaves inoculated with BSMV:PstGSRE4 compared with the leaves of mock- and BSMV:γ-inoculated wheat (**S7D Fig**). In addition, histological analysis by fluorescence microscopy revealed that initial haustorium formation and growth of secondary hyphae both were reduced in BMSV:PstGSRE4-1/2as infected plants (**S8A–S8C Fig**).

To further confirm the virulence function of *PstGSRE4* in *Pst* infection using stable transgenic plants, we prepared the RNA interference (RNAi) construct pAHC25-*PstGSRE4*-RNAi (**S9A Fig**) and delivered it into wheat cv. XN1376 by particle bombardment. Two $T_4$ transgenic wheat lines (L19 and L76) containing pAHC25-*PstGSRE4*-RNAi displayed significantly enhanced resistance against the virulent *Pst* race CYR31 (**Fig 1A**). Transcript levels of *PstGSRE4* during *Pst* infection of transgenic lines L19 and L76 were significantly reduced (**Fig 1B**). The results showed that both lines L19 and L76 contained at least one copy of the transgene (**S9B Fig**). Compared with control plants, fungal biomass in infected leaves of transgenic lines L19 and L76 was significantly reduced (**Fig 1C**). We also detected the contents of $H_2O_2$ and $O_2^-$ in infected leaves of transgenic lines L19 and L76 at 6, 12, 24 and 48 hpi (**Figs 1D and S9E**). The result showed that the accumulation of $H_2O_2$ was significantly increased and the accumulation of $O_2^-$ was significantly reduced in transgenic lines L19 and L76, suggesting that PstGSRE4 regulates the accumulation of $H_2O_2$ in wheat. The formation of mature haustoria and the development of secondary hyphae were assessed by microscopy. The results showed

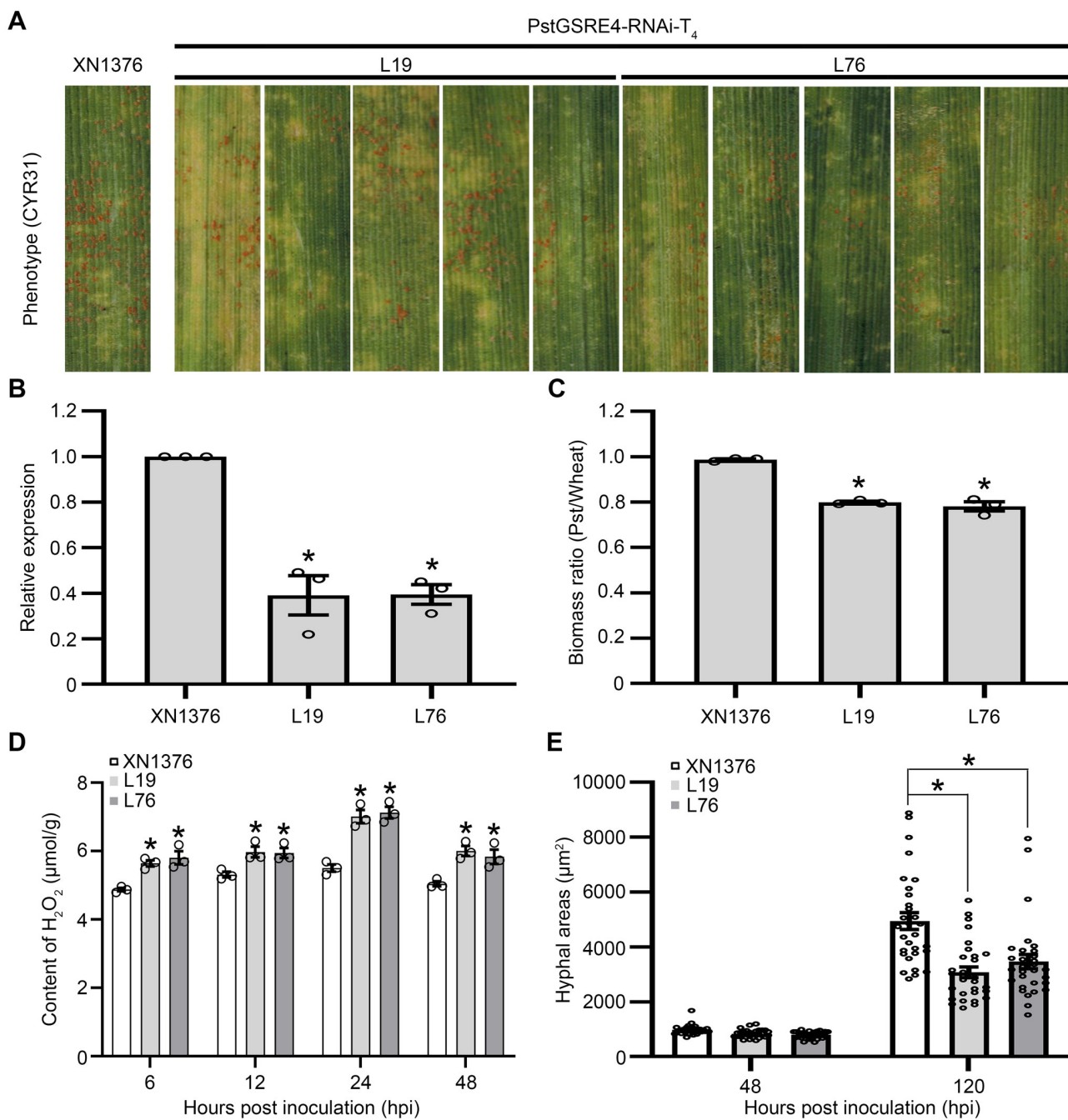

**Fig 1. RNAi-mediated stable silencing of *PstGSRE4* significantly reduces Virulence of *Pst* and increases H₂O₂ accumulation in Wheat. (A)** Phenotypes of the second leaves of the fourth generation of wheat plants at 14 dpi with *Pst*. **(B)** Relative transcript levels of *PstGSRE4* in different transgenic lines at 48 hpi with *Pst*. Values represent the means ± SE (n = 3). **(C)** Ratio of fungal to wheat nuclear content using fungal *PstEF-1* and wheat *TaEF-1α* genes, respectively. Genomic DNA was extracted from the second leaf from three different plants at 14 dpi. Values represent the means ± SE (n = 3). **(D)** Content of H₂O₂ in different transgenic lines at 6, 12, 24 and 48 hpi. Values represent the means ± SE of three independent samples. **(E)** Quantification of hyphal areas in different transgenic lines at 48 and 120 hpi. Values represent the means ± SE (n = 30). Differences were assessed using Student's *t*-test. Asterisk indicates $P < 0.05$.

that hyphal areas during infection of L19 and L76 were significantly reduced at 120 hpi (**Figs 1E** and **S9C**). $H_2O_2$ accumulation was also measured during *Pst* infection of transgenic plants at 72 and 120 hpi. We found that $H_2O_2$ accumulation was significantly increased in the transgenic plants relative to the control (**S9D and S9F Fig**). These results indicated that *PstGSRE4* contributes to virulence of *Pst* on wheat leaves.

## Overexpression of *PstGSRE4* reduces wheat resistance to *Pst*

To further confirm the virulence function of *PstGSRE4* in *Pst* infection in stable transgenic plants, we prepared the overexpression construct pCAMBIA3301-*PstGSRE4*-overexpression (**S10A Fig**) and delivered it into wheat cv. Fielder by *A. tumefaciens*-mediated stable transformation. Two $T_3$ transgenic lines (L2 and L3) containing pCAMBIA3301-*PstGSRE4*-overexpression displayed significantly reduced wheat resistance against *Pst* race CYR23 (**Fig 2A**). The results showed that both L2 and L3 contained at least one copy of the transgene (**S10B and S10C Fig**). RT-PCR (**S10D Fig**) and qRT-PCR (**Fig 2B**) showed that the expression of *PstGSRE4* was different in each line, and it influenced the resistance against *Pst* race CYR23 to a different extent. Compared with control plants, fungal biomass in infected leaves of transgenic lines L2 and L3 was significantly increased (**Fig 2C**). We also detected the contents of $H_2O_2$ and $O_2^-$ in infected leaves of transgenic lines L2 and L3 at 6, 12, 24 and 48 hpi (**Figs 2D** and **S10E**). The result showed that the accumulation of $H_2O_2$ was significantly reduced and the accumulation of $O_2^-$ was significantly increased in transgenic lines L2 and L3, suggesting that PstGSRE4 regulates the accumulation of $H_2O_2$ in wheat. As shown in **Figs 2E** and **S10F–S10H**, $H_2O_2$ accumulation and HR triggered by *Pst* race CYR23 were significantly decreased in *PstGSRE4*-overexpression plants at 48 hpi. These findings indicate that *PstGSRE4* is an important pathogenicity factor that regulates the accumulation of $H_2O_2$ during *Pst* infection.

## PstGSRE4 specifically targets wheat copper zinc superoxide dismutase TaCZSOD2

To understand the potential virulence function of PstGSRE4 in wheat, a yeast two-hybrid (Y2H) library was constructed to screen constructs and identify potential host targets of PstGSRE4. With PstGSRE4 (ΔSP) as the bait, several candidate targets were identified (**S3 Table**). We selected the candidate ROS-associated genes for further study. A candidate target sequence was annotated as superoxide dismutase [Cu-Zn] (TraesCS7A02G292100.1) (http://plants.ensembl.org/index.html). According to a previous study, we collected 26 TaSODs, 8 AtSODs and 8 OsSODs from the Arabidopsis Information Resource (TAIR10) database (http://www.arabidopsis.org/index.jsp), the Rice Genome Annotation Project (RGAP) database (http://rice.plantbiology.msu.edu/) and the wheat reference genome IWGSC v1.1 (E-value $< 1e^{-5}$), respectively. A phylogenetic tree was constructed, which revealed that the CuZuSOD gene obtained in this study lies within the same clade as AtCSD2 (**S11 Fig** and **S1 File**). Based on this evidence, we designated the wheat CuZnSOD as TaCZSOD2. BlastN analyses of the wheat genome showed that there were three copies of TaCZOD2 located on chromosomes 7A, 7B and 7D, respectively. Sequence alignment showed that the three copies, designated as TaCZSOD2-7A, TaCZSOD2-7B and TaCZSOD2-7D, and the TaCZSOD2 obtained in this study share 99.3% nucleotide identity (**S12 Fig**). We cloned TaCZSOD2 and the other three TaCZSODs (TaCZSOD1, TaCZSOD3, TaCZSOD4) from wheat cultivar Suwon11 to further confirm the preliminary results of the Y2H assay (**Fig 3A and 3B**). The results showed that PstGSRE4 specifically interacts with TaCZSOD2.

To determine whether PstGSRE4 can directly interact with TaCZSOD2, we used recombinant proteins PstGSRE4-GST, TaCZSOD2-His and GST, TaCZSOD1-His (as negative

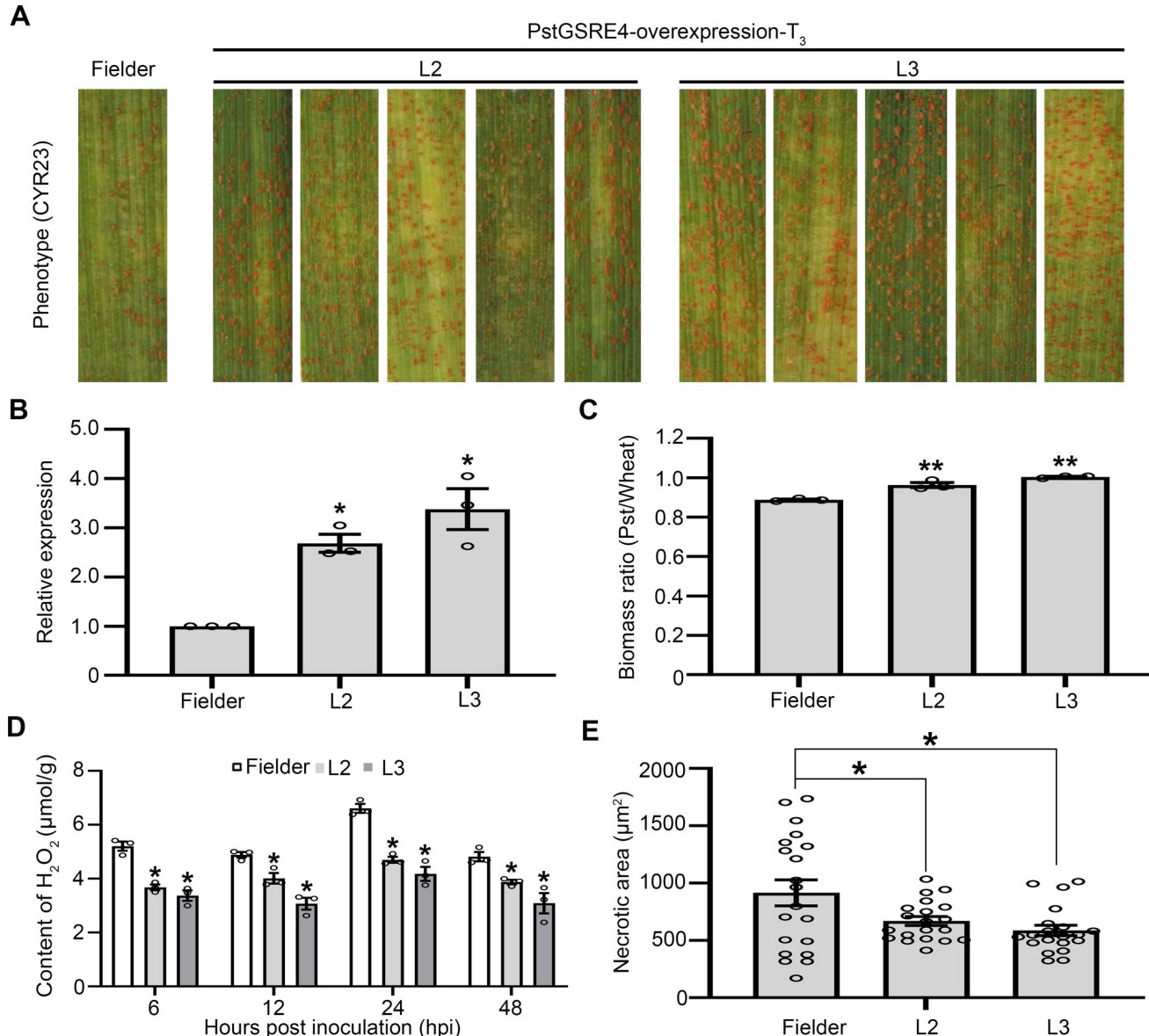

**Fig 2. Overexpression of *PstGSRE4* significantly increases virulence of *Pst* and reduces H₂O₂ accumulation in wheat.** (A) Phenotypes of the second leaves of the third generation of wheat plants at 14 dpi with *Pst*. (B) Relative transcript levels of *PstGSRE4* in different transgenic lines at 48 hpi with *Pst*. Values represent the means ± SE (n = 3). (C) Ratio of fungal to wheat nuclear content determined using the contents of fungal *PstEF1* and wheat *TaEF-1α* genes, respectively. Genomic DNA was extracted from the second leaf from three different plants at 14 dpi. Values represent the means ± SE (n = 3). (D) Content of H₂O₂ in different transgenic lines at 6, 12, 24 and 48 hpi. Values represent the means ± SE of three independent samples. (E) Quantification of necrotic cell death area in different transgenic lines at 48 hpi. Values represent the means ± SE (n = 20). Differences were assessed using Student's *t*-test. Asterisk indicates *P* < 0.05, and double asterisk indicates *P* < 0.01.

control) expressed from *E. coli* BL21 to conduct GST pull-down assay (**Fig 3C**). We detected immunoprecipitated protein complexes by western blotting. TaCZSOD2-His was detected in PstGSRE4-GST pull-down fractions but not the TaCZSOD1-His, indicating that PstGSRE4 specifically interacts with TaCZSOD2 *in vitro*. To obtain further experimental evidence, we conducted a Co-IP experiment based on *A. tumefaciens*-mediated transient expression in *N. benthamiana*. PstGSRE4 mature protein fused to GFP and TaCZSOD2 or TaCZSOD1 fused to HA were co-expressed in *N. benthamiana*. Western blotting analysis showed that PstGSRE4

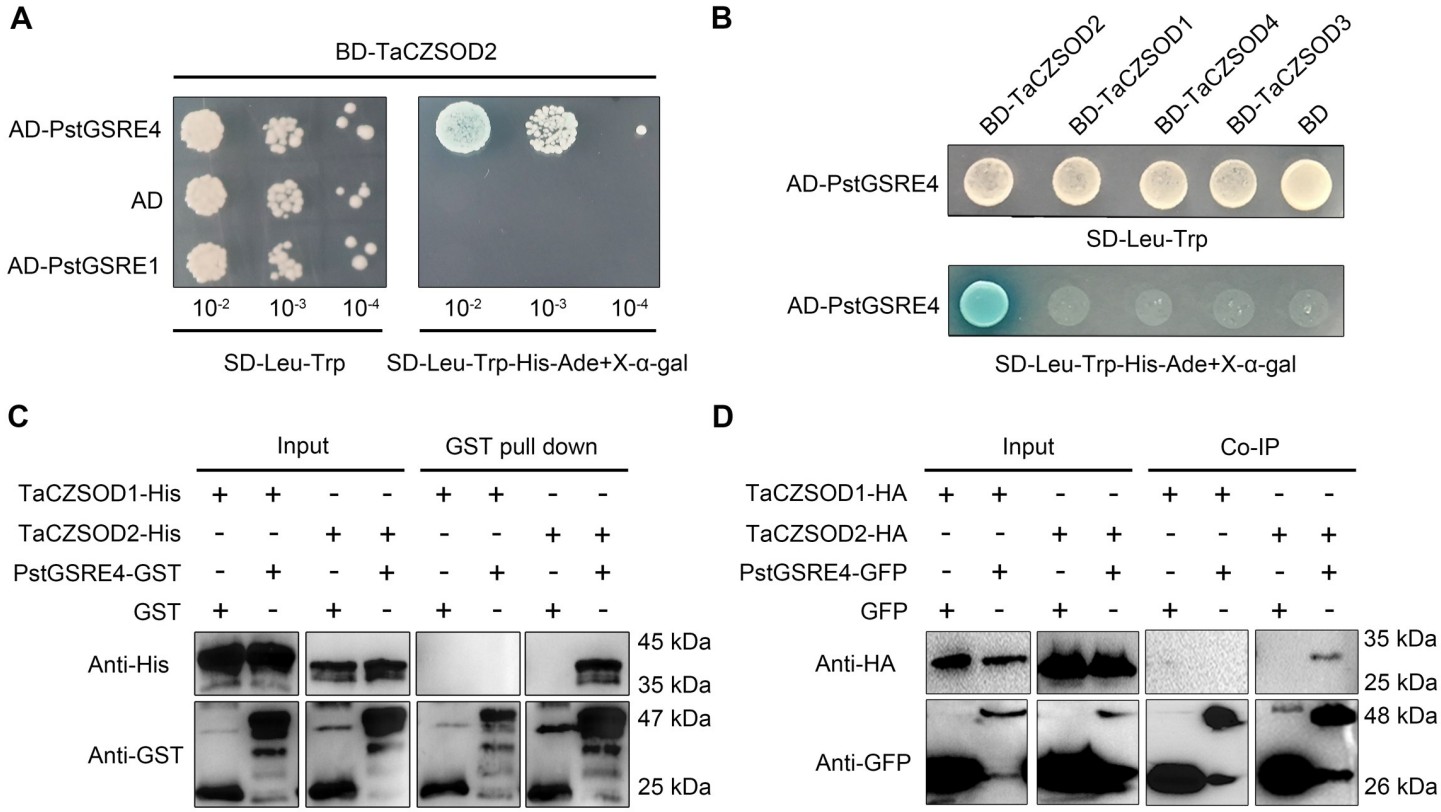

**Fig 3. PstGSRE4 specifically interacts with wheat TaCZSOD2 *in vitro* and *in vivo*. (A)** PstGSRE4 interacts with TaCZSOD2 in yeast. Only the yeast co-expressing PstGSRE4 and TaCZSOD2 grew on the medium SD-Trp-Leu-His-Ade and yielded X-α-gal activity. Co-expressing PstGSRE1 and TaCZSOD2 cannot grow on the medium SD-Trp-Leu-His-Ade. **(B)** PstGSRE4 specifically interacts with TaCZSOD2 in yeast. Only the yeast co-expressing PstGSRE4 and TaCZSOD2 grew on the medium SD-Trp-Leu-His-Ade and yielded X-α-gal activity. Co-expressing PstGSRE4 and TaCZSOD1, TaCZSOD3 or TaCZSOD4 cannot grow on the medium SD-Trp-Leu-His-Ade and yielded X-α-gal activity. **(C)** PstGSRE4 interacts with TaCZSOD2 *in vitro*. A GST pull-down assay was used to detect the interaction between PstGSRE4-GST and TaCZSOD2-His. TaCZSOD2-His and PstGSRE4-GST were detected with anti-His or anti-GST antibodies, respectively. TaCZSOD1-His was used as negative control. **(D)** PstGSRE4 interacts with TaCZSOD2 *in vivo*. Co-immunoprecipitation (IP) was performed on extracts of *N. benthamiana* leaves expressing both PstGSRE4-GFP and TaCZSOD2-HA. GFP was detected by western blot with anti-GFP antibodies. HA was detected by western blot with anti-HA antibodies. The TaCZSOD1-HA was used as negative control.

specifically interacts with TaCZSOD2 *in vivo* (**Fig 3D**). We also co-expressed PstGSRE4 and TaCZSOD2 in *N. benthamiana* cells and found that PstGSRE4 and TaCZSOD2 were co-localized in the cytoplasm (**S13A and S13B Fig**), suggesting that PstGSRE4 and TaCZSOD2 interact in the cytoplasm. To further confirm that PstGSRE4 and TaCZSOD2 interact in the cytoplasm, coding sequences of ΔTP-*TaCZSOD2* (*TaCZSOD2* without chloroplast transit peptide) was ligated into pBINGFP2 to construct the recombinant plasmid ΔTP-TaCZSOD2-GFP. And we co-expressed ΔTP-TaCZSOD2-GFP with PstGSRE4-RFP, TaCZSOD2-GFP with PstGSRE4-RFP, ΔTP-TaCZSOD2-GFP with RFP, and TaCZSOD2-GFP with RFP in *N. benthamiana*, and detected the activity of CuZnSOD. The results showed that TaCZSOD2 had enzyme activity in the cytoplasm, while PstGSRE4 inhibited the activity of TaCZSOD2 in the cytoplasm (**S13C and S13D Fig**). Compared with full length-TaCZSOD2, PstGSRE4 significantly inhibited the activity of ΔTP-TaCZSOD2 (**S13C and S13D Fig**).

## TaCZSOD2 positively regulates wheat resistance against *Pst*

To confirm the function of *TaCZSOD2* in wheat resistance to *Pst*, we knocked down expression of *TaCZSOD2* in wheat leaves by BSMV-VIGS. Two specific fragments were designed for

silencing *TaCZSOD2* (**S14A Fig**). qRT-PCR analysis showed that, during the interaction between wheat and *Pst*, transcript levels of *TaCZSOD2* were up-regulated at 12, 48 and 96 hpi with the avirulent *Pst* race CYR23, and up-regulated at 96 and 120 hpi with the virulent *Pst* race CYR31 (**S14B Fig**). After virus inoculation, plants displayed mild chlorotic mosaic symptoms at 10 dpi. Compared with the control plants, fewer necrotic spots and sporadic uredia appeared on leaves of *TaCZSOD2*-knockdown plants after inoculation with the avirulent race CYR23, whereas no significant differences were found on leaves of *TaCZSOD2*-knockdown plants after inoculation with the virulent race CYR31 (**Fig 4A**). The silencing efficiency monitored by qRT-PCR indicated that *TaCZSOD2-1as/2as* transcript levels in knockdown plants were significantly reduced (**Fig 4B**). Moreover, the transcript levels of the other *TaCZSODs* were not influenced after the expression of *TaCZSOD2* was knocked down (**Fig 4C**). The biomass of *Pst*/wheat showed an increase in the leaves inoculated with BSMV:TaCZSOD2 after inoculation with the avirulent race CYR23 compared with the leaves of BSMV:γ-inoculated wheat (**Fig 4D**). The biomass of *Pst*/wheat showed no significant changes in the leaves inoculated with BSMV:TaCZSOD2 after inoculation with the virulent race CYR31 compared with the leaves of BSMV:γ-inoculated wheat (**S14C Fig**). We detected the activity of CuZnSOD in *TaCZSOD2*-knockdown plants after inoculation with the avirulent race CYR23. Compared with the control plants, the enzyme activity of *TaCZSOD2*-knockdown plants was significantly reduced (**Fig 4E**). We also detected the contents of $H_2O_2$ and $O_2^-$ in infected leaves of *TaCZSOD2*-knockdown plants at 6, 12, 24 and 48 hpi (**Figs 4F and S14D**). The results showed that knockdown of *TaCZSOD2* significantly reduced $H_2O_2$ accumulation and increased $O_2^-$ accumulation in wheat. Histological changes in *TaCZSOD2*-knockdown plants infected with CYR23 were observed by microscopy. As shown in **Figs 4G** and **S14E–S14G**, $H_2O_2$ accumulation and HR triggered by the avirulent *Pst* race CYR23 were significantly decreased in *TaCZSOD2*-knockdown plants at 48 hpi.

To further test the function of *TaCZSOD2* in wheat defense against *Pst*, we also performed transient delivery of *TaCZSOD2* using the bacterial T3SS system in wheat. At 48 h post-infiltration with a plasmid carrying EtHAn strains, wheat plants were inoculated with *Pst* virulent race CYR31. The number of uredia was significantly reduced at 14 dpi (**S15A Fig**). In addition, the area of $H_2O_2$ accumulation and HR areas were significantly increased (**S15B–S15D Fig**).

To further confirm the function of *TaCZSOD2* during *Pst* infection of stable transgenic plants, we prepared the overexpression construct CUB-*TaCZSOD2*-overexpression (**S16A Fig**) and delivered it into wheat cv. Fielder by *A. tumefaciens*-mediated stable transformation. Three $T_2$ transgenic lines (L1, L4 and L9) containing CUB-*TaCZSOD2*-overexpression displayed significantly increased wheat resistance against *Pst* race CYR31 (**Fig 5A**). PCR and Western blot assays indicated that L1, L4 and L9 contained at least one copy of the transgene (**S16B and S16C Fig**). qRT-PCR showed that the expression of *TaCZSOD2* increased in each line (**S16D Fig**). And the fungal biomass in infected leaves of transgenic lines L1, L4 and L9 were significantly reduced (**Fig 5B**). Also, compared with control plants, the enzyme activities of CuZnSOD in transgenic lines L1, L4 and L9 were significantly increased (**Fig 5C**). We also detected the contents of $H_2O_2$ and $O_2^-$ in infected leaves of *TaCZSOD2*-overexpression plants at 6, 12, 24 and 48 hpi (**Figs 5D and S16E**). The results showed that overexpression of *TaCZSOD2* significantly increased $H_2O_2$ accumulation and reduced $O_2^-$ accumulation in wheat. Histological changes in *TaCZSOD2*-overexpression plants infected with CYR31 were observed by microscopy. As shown in **Figs 5E** and **S16F–S16H**, $H_2O_2$ accumulation and HR area triggered by *Pst* race CYR31 were significantly increased in *TaCZSOD2*-overexpression plants at 48 hpi/72 hpi with the increasing enzyme activity of CuZnSOD. These findings indicate that TaCZSOD2 positively regulates wheat resistance against *Pst*.

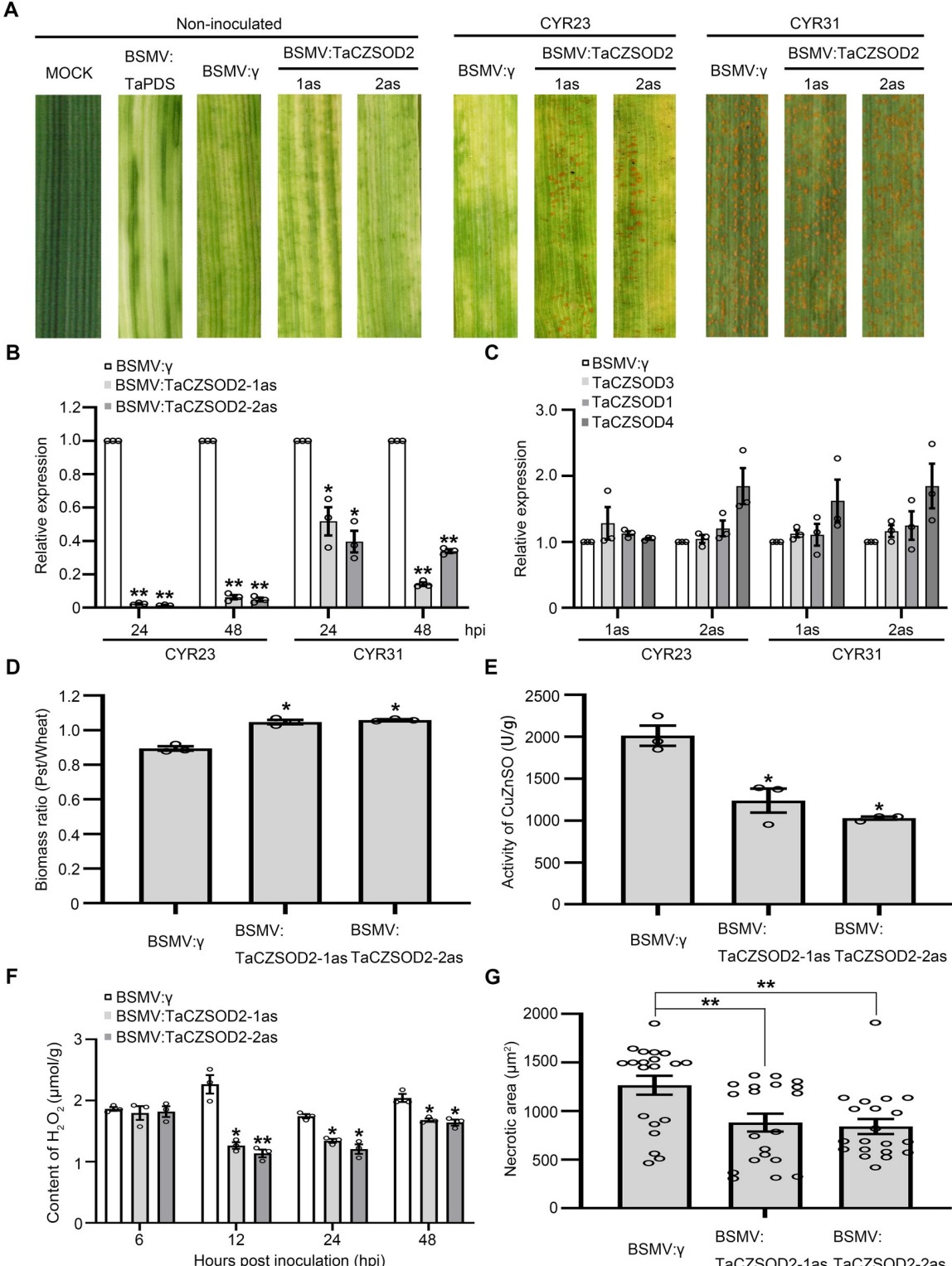

**Fig 4. *TaCZSOD2* is a positive regulator of wheat immunity.** (A) Phenotypes of the fourth leaves of knockdown plants inoculated with *Pst* race CYR23 and CYR31 at 12 dpi. (B) Relative transcript levels of *TaCZSOD2* in *TaCZSOD2*-knockdown plants challenged by CYR23 and CYR31. *TaEF-1α* was used for normalization. Values represent the means ± SE (n = 3). (C) Relative transcript levels of other *TaCZSODs* in *TaCZSOD2*-knockdown plants challenged by CYR23 at 48 hpi. *TaEF-1α* was used for normalization. Values represent the means ± SE (n = 3). (D) After inoculation with CYR23, ratio of fungal to wheat nuclear content using fungal *PstEF-1* and wheat *TaEF1α*

genes, respectively. Values represent the means ± SE (n = 3). **(E)** The enzyme activity of CuZnSOD in *TaCZSOD2*-knockdown plants. Values represent the means ± SE of three independent samples. **(F)** Content of $H_2O_2$ in *TaCZSOD2*-knockdown plants at 6, 12, 24 and 48 hpi. Values represent the means ± SE of three independent samples. **(G)** Quantification of necrotic cell death area in *TaCZSOD2*-knockdown plants at 48 hpi. Values represent the means ± SE (n = 20). Differences between time-course points were assessed using Student's *t*-tests. Asterisks indicate $P < 0.05$, double asterisks indicate $P < 0.01$.

## TaCZSOD2 increases $H_2O_2$ accumulation

In order to further confirm that TaCZSOD2 can increase $H_2O_2$ accumulation, we use chitin to induce a rapid oxidative burst to determine the capacity of TaCZSOD2 to maintain redox balance in *TaCZSOD2*-knockdown plants and *TaCZSOD2*-overexpression transgenic plants. qRT-PCR analysis showed that the transcript levels of *TaCZSOD2* in knockdown plants were significantly reduced (**S17A Fig**) and next we found that the chitin-induced $H_2O_2$ accumulation in *TaCZSOD2*-knockdown plants were markedly reduced compared with the wild-type (**S17B Fig**). Meanwhile, qRT-PCR analysis showed that the transcript levels of *TaCZSOD2* in transgene plants were significantly increased (**S17C Fig**) and the chitin-induced $H_2O_2$ accumulation in *TaCZSOD2*-overexpression transgenic plants were markedly increased compared with the wild-type (**S17D Fig**). In summary, these results suggested that TaCZSOD2 has the ability to increase $H_2O_2$ accumulation.

## PstGSRE4 inhibits the activity of TaCZSOD2

In order to confirm the effect of PstGSRE4 on the activity of TaCZSOD2, we used recombinant proteins PstGSRE4-His, TaCZSOD2-His and eGFP-His, TaCZSOD1-His (as negative control) expressed in *E. coli* BL21 to conduct enzyme activity assays by the NBT photoreduction method (**Fig 6A and 6B**). The results demonstrated that PstGSRE4 reduces the enzyme activity of TaCZSOD2 *in vitro*, but not TaCZSOD1. Meanwhile, we used agro-infiltration to transiently co-express PstGSRE4 and TaCZSODs in *N. benthamiana* to determine whether PstGSRE4 inhibits the activity of TaCZSOD2 (**Fig 6C and 6D**). The results showed that PstGSRE4 reduces the activity of TaCZSOD2 *in vivo*. To further explore the effect of PstGSRE4 on TaCZSOD2 *in vivo*, we also used *PstGSRE4*-overexpression transgenic lines L2 and L3 to detect the activity of CuZnSOD in wheat (**Fig 6E and 6F**). The results showed that the activity of CuZnSOD was reduced when the expression of *PstGSRE4* increased. These results indicated that PstGSRE4 can reduce the enzyme activity of TaCZSOD2.

## Discussion

Glycine- or serine-rich proteins perform important, even decisive roles during infection in various pathogens [27–29,30–33]. However, few glycine-serine-rich effectors (GSREs) have been characterized in *Pst*. In our previous study, we identified four glycine-serine-rich effectors in *Pst*. Among these candidates, we focused on PstGSRE1, which contains the m9 motif, and was found to target the ROS-associated transcription factor TaLOL2 [29]. Further sequence analysis of the GSREs revealed that only PstGSRE4 lacks the m9-like motif (**S1B Fig**), suggesting the functional divergence with other GSREs. Silencing *PstGSRE4* decreased *Pst* growth and development (**Figs 1 and S7 and S8**), due to the increased ROS accumulation in wheat. ROS have been proposed to orchestrate the establishment of plant defenses following HR [3,34]. In plant cells, ROS has been identified as playing a key role in the development of HR and systemic immunity [35]. Inhibition of this reaction is an important strategy for the successful infection and colonization by obligate biotrophic pathogens. Overexpression of *PstGSRE4* suppressed ROS accumulation induced by the avirulent *Pst* race CYR23 and the deposition of callose induced by EtHAn (**S6 Fig**), which promoted the infection by *Pst*. Like

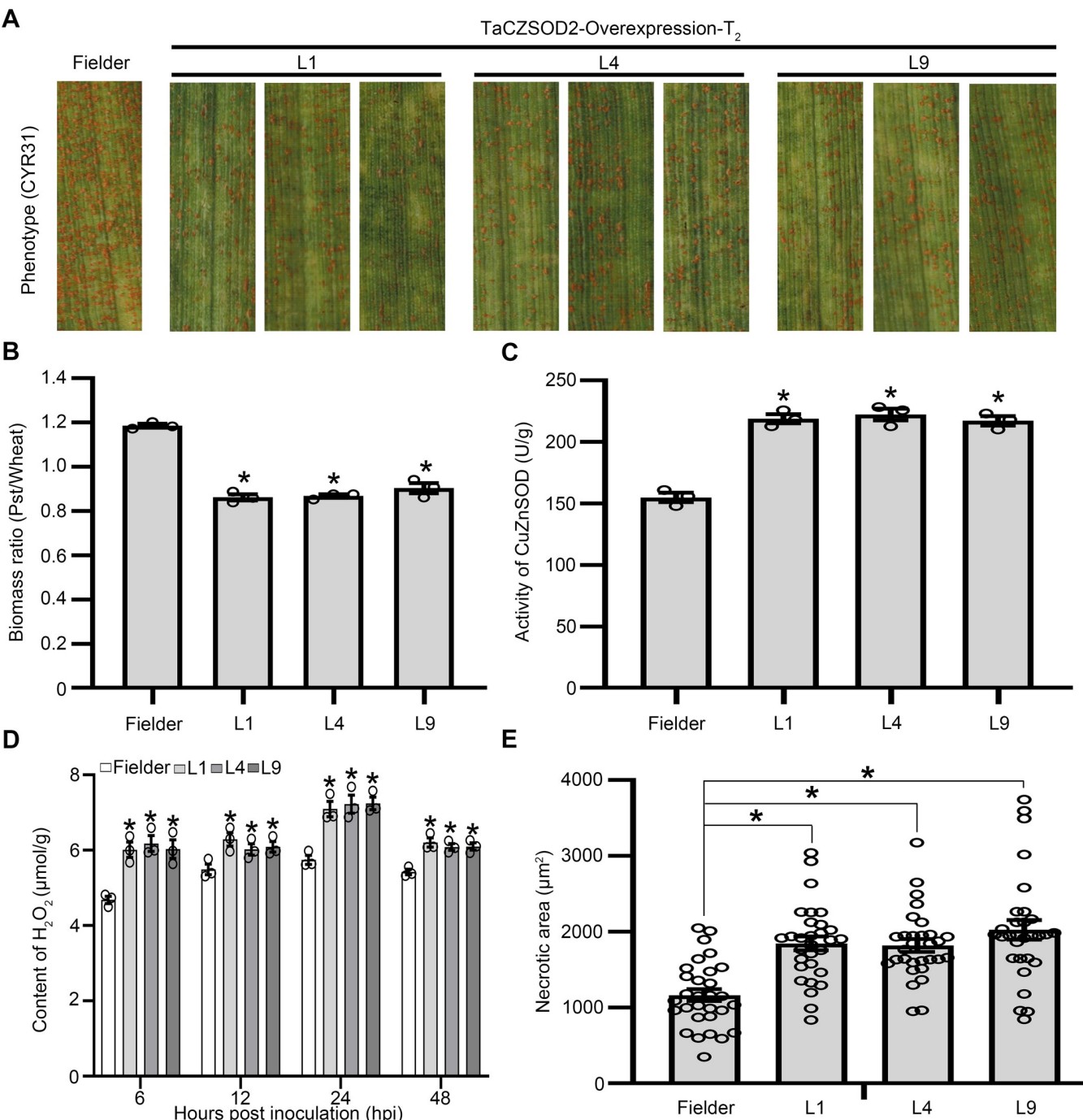

**Fig 5. Overexpression of *TaCZSOD2* significantly increases H₂O₂ accumulation and confers the resistance of wheat to *Pst*. (A)** Phenotypes of the third leaves of the second generation of wheat plants at 14 dpi with *Pst*. **(B)** Ratio of fungal to wheat nuclear content determined using the contents of fungal *PstEF1* and wheat *TaEF-1α* genes, respectively. Genomic DNA was extracted from the second leaf from three different plants at 14 dpi. Values represent the means ± SE (n = 3). **(C)** The enzyme activity of CuZnSOD in *TaCZSOD2*-overexpression plants at 48 hpi with *Pst*. Values represent the means ± SE of three independent samples. **(D)** Content of H₂O₂ in different transgenic lines at 6, 12, 24 and 48 hpi. Values represent the means ± SE of three independent samples. **(E)** Quantification of necrotic cell death area in different transgenic lines at 72 hpi. Values represent the means ± SE (n = 30). Differences were assessed using Student's *t*-test. Asterisk indicates $P < 0.05$.

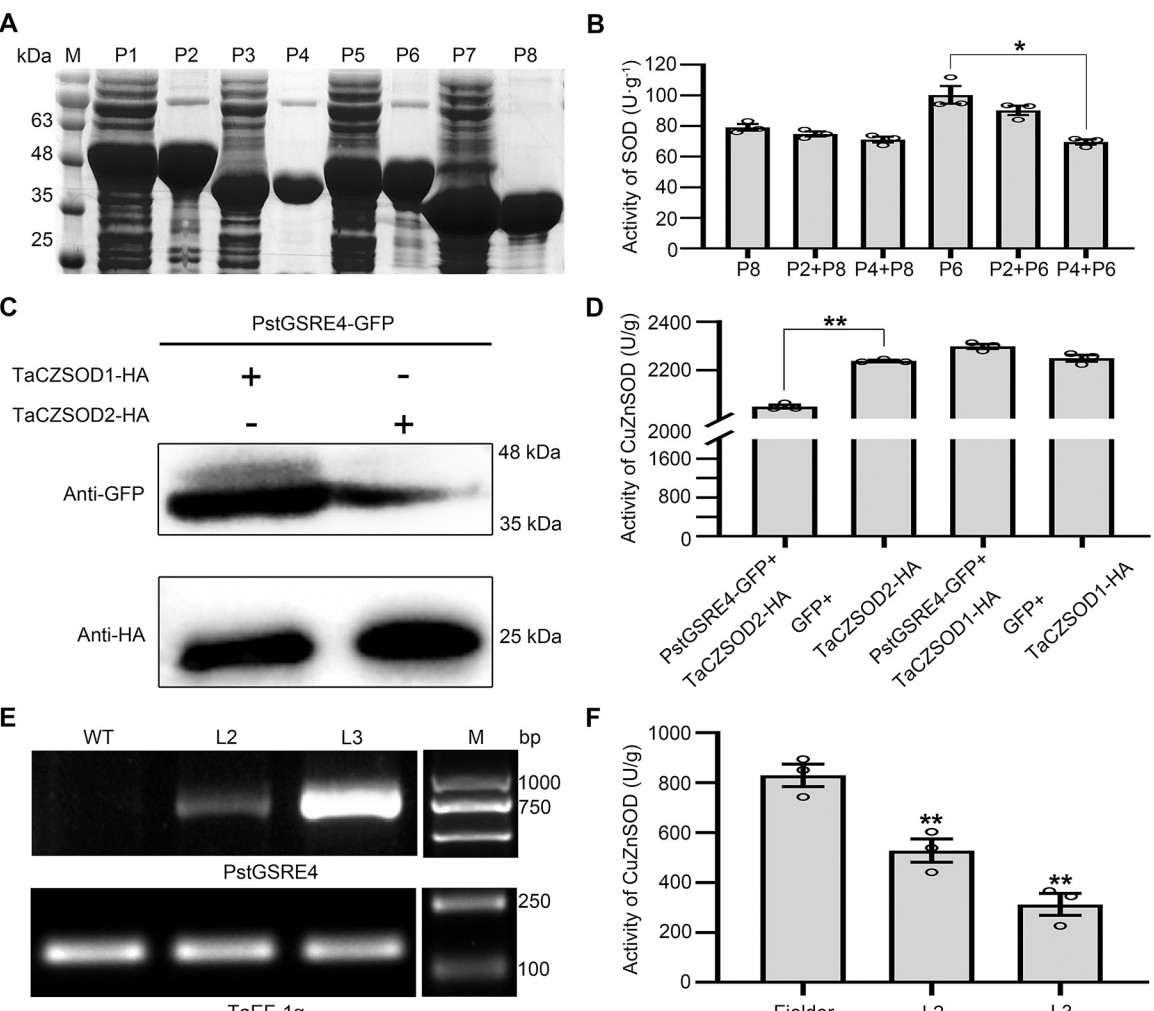

**Fig 6. PstGSRE4 inhibits the activity of TaCZSOD2 *in vitro* and *in vivo*. (A)** SDS-PAGE analysis of recombinant proteins purified from *E. coli*. The protein bands were visualized by Coomassie staining. M, Marker. P1, recombinant protein eGFP-His with induction by isopropyl-β-D-thiogalactopyranoside (IPTG). P2, purified recombinant protein eGFP-His. P3, recombinant protein PstGSRE4-His with induction by IPTG. P4, purified recombinant protein PstGSRE4-His. P5, recombinant protein TaCZSOD2-His with induction by IPTG. P6, purified recombinant protein TaCZSOD2-His. P7, recombinant protein TaCZSOD1-His with induction by IPTG. P8, purified recombinant protein TaCZSOD1-His. **(B)** Nitro-blue tetrazolium (NBT) photoreduction method was used to detect the activity of SOD. The TaCZSOD1-His was used as negative control. **(C-D)** Using the co-expression method to detect the activity of CuZnSOD in tobacco. Western blot was performed to show normal expression of PstGSRE4-GFP with anti-GFP antibody, TaCZSOD1-HA and TaCZSOD2-HA with anti-HA antibody. Co-express PstGSRE4-GFP and TaCZSOD1-HA/TaCZSOD2-HA respectively in tobacco, then detect the activity of CuZnSOD by using the CuZnSOD assay kit after 48 hpt (hours post treatment). **(E-F)** Using *PstGSRE4*-overexpression lines to detect the activity of CuZnSOD in wheat. The activity of CuZnSOD in different transgene lines were detected by using the CuZnSOD assay kit at 48 hpi with *Pst*. RT-PCR was used to detect the expression of *PstGSRE4*. Expression of *TaEF-1α* showed equal loading. WT, wild type, cv. Fielder. L2-L3, the third generation of *PstGSRE4*-overexpression wheat lines. Values represent the means ± SE of three independent samples. Asterisks indicate $P < 0.05$, double asterisks indicate $P < 0.01$.

PstGSRE4, *Pst* effector PstGSRE1 was also reported to be involved in suppression of callose deposition and ROS accumulation [29]. In *M. oryzae*, some glycine- or serine-rich effectors regulate the activity of a variety of antioxidant enzymes, then inhibit the level of ROS in the host, and finally decrease the host immune response [30,31]. Therefore, we speculated that GSRE proteins play an important role in regulating the host immune response, and they may specifically regulate the ROS signal transduction pathway of higher plants. However, although

PstGSRE4 contains 12.93% glycine and 19.40% serine, it does not interact with TaLOL2 (**S1 Fig**), suggesting that it regulates ROS signal pathway in wheat via a different mechanism.

In this study, we found that PstGSRE4 specially interacts with wheat copper zinc superoxide dismutase TaCZSOD2 (**Fig 3**), an important isoform of SOD in plants. The plant CuZnSOD isoenzymes differ in their subcellular location, either plastid or cytosolic. Distinct subcellular localization of the individual CuZnSOD isoforms indicates the necessity of superoxide removal within specialized cellular compartments [36]. In Arabidopsis, the three CuZnSOD isoforms (CSD1, CSD2, and CSD3) are localized in the cytoplasm and nucleus, chloroplast and peroxisome, respectively. However, in this study, we found that TaCZSOD2, the ortholog of CSD2 of *Arabidopsis*, is not only localized in the chloroplast, but also in the cytoplasm (**S13A Fig**). Overexpression of *ΔTP-TaCZSOD2* in *N. benthamiana* improved the CuZnSOD enzyme activity, but slightly lower than overexpression of full length of *TaCZSOD2* (**S13C and S13D Fig**), suggesting that TaCZSOD2 of the chloroplast can also be activated via the CCS-independent pathway when localized in cytoplasm like CSD2 in *Arabidopsis* [37]. Meanwhile, PstGSRE4 inhibits the CuZnSOD enzyme activity that is increased by overexpression of *TaCZSOD2* or *ΔTP-TaCZSOD2* in *N. benthamiana* (**S13C and S13D Fig**). Thus, our data suggest that TaCZSOD2 can simultaneously carry out the function of the enzyme in the cytoplasm and chloroplast. The mechanism by which PstGSRE4 interacts with TaCZSOD2 in the cytoplasm remains to be further investigated. In addition, PstGSRE4 interacts only with TaCZSOD2, suggesting that there is a specific recognition site between PstGSRE4 and TaCZSOD2, but that site remains to be identified. Moreover, like rice CSD2 [21], silencing of Ta*CZSOD2* resulted in reduced CuZnSOD enzyme activity (**Fig 4E**), whereas overexpression of *TaCZSOD2* led to higher CuZnSOD enzyme activity (**Fig 5C**), suggesting that TaCZSOD2 is an important CuZnSOD enzyme and positively contributes to CuZnSOD activity. In the process of plant-pathogen interaction, the activity of superoxide dismutase (SOD) positively or negatively correlates with plant disease resistance [38–41], this may be because each pathogen has its own unique mechanism of pathogenicity. However, no reports define the function of CuZnSOD in the interaction between wheat and rust fungi. In our study, silencing *TaCZSOD2* in wheat reduced the $H_2O_2$ accumulation triggered by the avirulent race CYR23 (**Figs 4F and S14E and S14F**). Meanwhile, compared with the control plants there were fewer necrotic spots and only sporadic uredia on *TaCZSOD2*-knockdown plants (**Fig 4A**). Moreover, bacterial delivery of *TaCZSOD2* into wheat tissue (**S15B and S15D Fig**) and *TaCZSOD2*-overexpression lines (**Figs 5D and S16F and S16H**) increased wheat resistance to *Pst* in a ROS-dependent manner. Thus, our results indicated that *TaCZSOD2* is a positive regulator of wheat resistance to *Pst* by increasing the accumulation of $H_2O_2$ in wheat.

$H_2O_2$ acts as a signal molecule to trigger resistance to various biotic and abiotic stresses [42–44]. Therefore, the pathogens usually secrete effectors to inhibit $H_2O_2$ accumulation [45,46]. Moreover, $H_2O_2$ also functions as intracellular and intercellular signal molecules to amplify the cellular ROS signals and trigger the HR [47–50]. HR is very effective against obligate biotrophic pathogens. During the interaction between wheat and *Pst*, the early burst of $O_2^-$ may be induced by the contact of surface structures of the haustorial initials with the plasma membrane of mesophyll cells, whereas $H_2O_2$ is induced to activate HR and other resistance responses [6]. In our study, PstGSRE4 inhibited the activity of TaCZSOD2 *in vitro* and *in vivo* (**Fig 6**), a detrimental response for wheat to accumulate more $H_2O_2$ and successfully resist *Pst*. Meanwhile, the conclusion is confirmed again by the results that $H_2O_2$ and HR induced by CYR23 were reduced in PstGSRE4-overexpression transgenic lines (**Figs 2D and 2E, and S10F–S10H**) and in *TaCZSOD2*-knockdown plants (**Figs 4F and 4G, and S14D–S14G**), along with the increase of $O_2^-$ level. Thus, our results proved that PstGSRE4 inhibits the activity of total CuZnSOD by inhibiting the enzyme activity of TaCZSOD2 to control moderate $H_2O_2$ accumulation upon *Pst* infection, and further promote stripe rust disease (**Fig 7**).

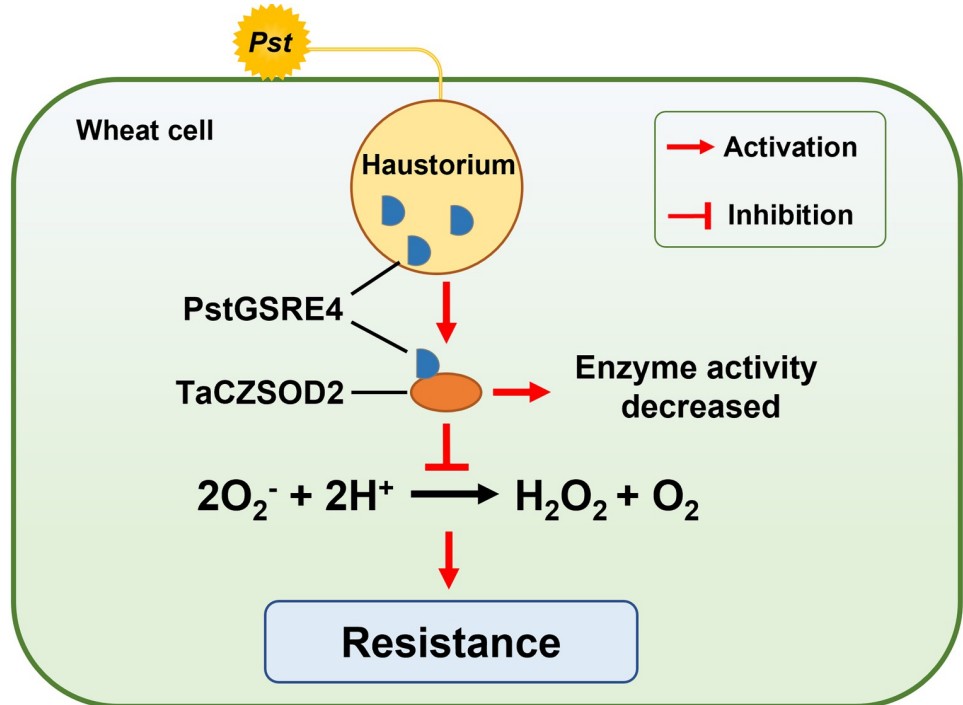

**Fig 7. A working model illustrating how PstGSRE4 suppresses ROS-associated wheat immunity by decreasing the activity of TaCZSOD2.** During infection, *Pst* effector PstGSRE4 is secreted and translocated into host cells. It targets TaCZSOD2 and inhibits the enzyme activity of TaCZSOD2, preventing TaCZSOD2 from carrying out its normal role of adjusting the ROS-mediated HR and disease resistance during the interaction between wheat and *Pst*.

In this study, we also identified other three highly conversed TaCZSODs, TaCZSOD1, TaCZSOD3, and TaCZSOD4. The enzyme activity of TaCZSOD1 has been also detected *in vitro* (**Fig 6A and 6B**), suggesting that TaCZSOD1 may catalyze the production of $H_2O_2$ from $O_2^-$ during *Pst* infection. Future work will be performed to investigate the functions of the three TaCZSODs during wheat-*Pst* interaction and reveal the molecular mechanisms of other *Pst* effectors targeting the CuZnSODs involved in wheat immunity.

In summary, we identified a new glycine-serine-rich effector protein lacking the m9-like motif, PstGSRE4, which inhibits the enzyme activity of the wheat TaCZSOD2 to modulate ROS-associated defense responses. To our knowledge, this is the first direct evidence demonstrating that an effector in phytopathogens regulates the activity of CuZnSOD isoenzymes to suppress plant immunity. Previous studies indicated that *CSD* genes (mainly *CSD1* and *CSD2*) from wheat and rice also play a positive role during abiotic stress [51,52]. Future studies will focus on using gene editing or overexpression technology to obtain broad-spectrum disease resistance and abiotic stress tolerance materials to advance novel strategies for protecting the wheat crop.

## Materials and methods

### Plant materials and fungal Strains

In this study, we used wheat cultivar Suwon11 (Su11), Fielder and *N. benthamiana*. Wheat cultivar Su11 is highly susceptible to CYR31 and CYR32 and highly resistant to CYR23 [53]. Wheat seedlings were planted, inoculated with *Pst* and maintained in accordance with the procedures and conditions described previously [54]. Wheat cultivar Fielder was used for

transgenic transformation. Co-immunoprecipitation (CoIP) analysis was performed in four-leaf tobacco seedlings permeated with agrobacterium tumefaciens GV3101.

Freshly collected urediniospores of *Pst* race CYR23 were obtained from the leaves of wheat cultivar Thatcher (MX169), while CYR31 and CYR32 were obtained from Suwon11. The wheat cultivars Su11 and MX169 were grown at 16°C in an artificial climate chamber.

## Plasmid construction

The *PstGSRE4* gene was cloned using complementary DNA from *Pst* CYR32. Full-length *TaCZSOD* genes were cloned from Su11. The amplicons were prepared using the appropriate restriction enzymes (**S4 Table**) and ligated into pBINGFP2 (a plasmid containing green fluorescent protein, GFP) for transient expression in tobacco, and pEDV6 for transient expression in wheat as well as pSUC2T7M13ORI (pSUC2), pEGX4T-1-GST, pET32a-His with the ClonExpress II One Step Cloning Kit (Vazyme Biotechnology, Nanjing, China). For VIGS analysis, specific cDNA segments of *PstGSRE4* and *TaCZSODs* were predicted by siRNA finder software Si-Fi and then inserted into BSMV-γ carriers with NotI and PacI restriction sites [55]. For Co-IP assays, coding sequences of *PstGSRE4* were ligated into pBINGFP2, and *TaCZSOD2* and *TaCZSOD1* were ligated into pICH86988 (a plasmid containing HA-tag), respectively. For Y2H assays, the coding sequences of *PstGSRE4* and *TaCZSODs* were separately prepared using appropriate restriction enzymes (**S4 Table**) and ligated into pGADT7 and pGBKT7 vectors. The *PstGSRE4* was ligated into pCAMBIA3301 for overexpression in wheat by *Agrobacterium*-mediated transformation, and the *TaCZSOD2* was ligated into CUB for overexpression in wheat by *Agrobacterium*-mediated transformation.

## qRT-PCR analysis

To assay expression levels of PstGSRE4, urediniospores and leaves of wheat Suwon11 infected with CYR32 at 6, 12, 18, 24, 36, 48, 72, 120, 168, 216 and 264 h post-inoculation (hpi) were harvested. To analyze the transcript levels of TaCZSOD2, leaves of Su11 inoculated with CYR23 and CYR31 at 0, 6, 12, 24, 48, 72, 96, 120 hpi were sampled. RNA of all samples was extracted with the Quick RNA isolation Kit (Huayueyang Biotechnology, China, Beijing, 0416-50GK). Approximately 2 μg of the total RNA were also used for reverse transcription using RevertAid First Strand cDNA Synthesis Kit (MNI, K1622). qRT-PCR on a CFX Connect Real-Time System (Bio-Rad, Hercules, CA, USA) was performed in a 25-μl reaction mixture containing 12.5 μl of LightCycler SYBR Green I Master Mix, 2 μl of diluted cDNA (1:5), 8.9 μl of distilled H$_2$O, 0.8 μl of forward primer (10 mM) and 0.8 μl of reverse primer (10 mM). The primers used are listed in **S4 Table**. Real-time PCR data were analyzed by the comparative $2^{-\Delta\Delta CT}$ method to quantify relative gene expression [25]. The expression levels of PstGSRE4 and TaCZSOD2 were normalized to *PstEF1* and *TaEF-1α*, respectively. Each sample was analyzed in three biological replications, and each PCR analysis included three technical repeats. The statistical significance was evaluated by Student's *t*-test.

## Yeast signal sequence trap system

To validate the function of the predicted signal peptide of PstGSRE4, the yeast signal sequence trap system was used as described previously [56]. The predicted signal peptide sequence of *PstGSRE4* was cloned into vector pSUC2T7M13ORI (pSUC2) using the specific primers (**S4 Table**) and then transformed into the invertase mutant yeast strain YTK12 [57]. To test the secretion function of the recombinant plasmid, positive clones were selected from the CMD-W medium then transferred to YPRAA medium to determine whether the recombinant plasmid had secretory function. In addition, invertase enzymatic activity was detected by the

reduction of 2,3,5-triphenyltetrazolium chloride (TTC) to insoluble red colored 1,3,5-triphenylformazan (TPF) according to procedures and conditions described previously [58].

## Agrobacterium tumefaciens infiltration assays

The sequence encoding PstGSRE4 without the signal peptide (PstGSRE4(ΔSP)) was ligated into pGR107 carrier to construct the agrobacterium recombinant plasmid PVX-PstGSRE4-HA. The *Avr1b* gene from *Phytophthora sojae* and eGFP-HA were used as controls (**S4 Table**). *A. tumefaciens* cultures were prepared as described previously [59]. Resuspended *A. tumefaciens* cultures carrying each effector gene or eGFP at a final $OD_{600}$ of 0.2 and 10 mM $MgCl_2$ buffer were infiltrated into the leaves of 4-week-old *N. benthamiana* using a syringe without a needle. After 24 h, *A. tumefaciens* cultures for delivery of Bax or Pst322 at a final $OD_{600}$ of 0.2 were also infiltrated into the same site of *N. benthamiana* leaves. Expression of genes in all infiltration sites was detected by immunoblot three days after infiltration. Symptoms were monitored and recorded from 3 to 8 d after infiltration. Three independent biological replicates were conducted for each experiment.

## Yeast two hybrid (Y2H) assay

TaCZSODs was constructed into pGBKT7 as bait, while PstGSRE4(ΔSP) was constructed into pGADT7 as prey (**S4 Table**). Then they were co-transformed into yeast strain AH109, plated on SD-Trp-Leu and SD-Trp-Leu-His medium, and cultured at 30˚C for 3 to 5 d. The monoclonals grown on SD-Trp-Leu-His were selected and diluted with water, then the interactions were confirmed by growth on the SD-Trp-Leu-His-Ade medium containing X-α-gal.

## Bacterial T3SS-mediated overexpression in wheat plants

pEDV6-*PstGSRE4(ΔSP)*, pEDV6-*TaCZSOD2* were transformed into *P. fluorescens* strain EtHAn by electroporation. pEDV6-RFP was used as a control. Infiltration into wheat leaves was performed according to the method described previously [60]. The involvement in *Pst* pathogenicity or host defense response was tested by challenging the second leaves in pEDV6-PstGSRE4-inoculated wheat plants with *Pst* avirulent race CYR23 after 24 h. For determination of $H_2O_2$ measurements, according to the previously described method [61], the inoculated leaves were sampled at 24 and 48 hpi and determined by 3–3'diaminobenzidine (DAB) staining. To examine the suppression of callose deposition, pEDV6-, pEDV6-PstGSRE4- and pEDV6-RFP-inoculated wheat plants were sampled at 48 hpi. Leaf samples were stained with 0.05% aniline blue in 67 mM $K_2HPO_4$ (pH 9.0) overnight in darkness [29]. Leaves were rinsed in water and mounted in 50% glycerol and examined under an Olympus BX-53 fluorescence microscope (Olympus Corporation, Tokyo, Japan) using a DAPI filter. Images were acquired using a constant setting with 1000-ms exposure time. The number of callose deposits was quantified using ImageJ software [62].

## Glutathione S-transferase (GST) pull-down assay

PstGSRE4(ΔSP) and TaCZSOD2/TaCZSOD1 were separately ligated into pGEX-4T-1 and pET22b/pET32a through enzyme digestion and ligation. Vectors were transformed into *E. coli* BL21 cells for protein expression. The corresponding protein was expressed and purified according to the prokaryotic expression procedure. GST-pull down kit (Thermo, Shanghai, China, UB281159) was used to validate the protein interactions *in vitro*. Another protein was detected by Western blot analysis. Horseradish peroxidase (HRP)-conjugated anti GST-Tag

rabbit polyclonal antibody (Cwbiotech, cat. no. CW0144M) and HRP conjugated anti His-Tag mouse monoclonal antibody (Cwbiotech, cat. no. CW0285M) were used for Western blots.

## Co-immunoprecipitation assays

PstGSRE4(ΔSP) and TaCZSOD2/TaCZSOD1 were ligated into pBINGFP2 and pICH86988 carriers, respectively. In addition, agrobacterium-mediated transient gene expression technology was used to co-express the above combinations in *N. benthamiana*. At 48 h after agroinfiltration, 100 μL of co-injected leaf proteins were extracted as the control (Input). Twenty μL of GFP Trap beads were added to the remaining extracts and incubated for 1 h, and centrifuged at 12000g at 4˚C for 1min. After removal of the supernatant, the beads in 60 uL volume of wash buffer were mixed with 20 uL of loading buffer, and heated at 100˚C for 5min. Precipitated proteins and crude proteins (Input) were detected by immunoblotting with an anti-GFP antibody (#A02020; Abbkine, Wuhan, China) and an anti-HA antibody (Beyotime, AF5057).

## Activity assays of CuZnSOD

PstGSRE4-15bs, TaCZSOD2-15bs/TaCZSOD1-15bs and GFP-15bs were expressed *in vitro* by a prokaryotic expression system, then diluted to the same concentration after purifying by His-tag Purification Resin (BeyoGold, P2210) according to the protocol of manufacturer. The activity of TaCZSOD2/TaCZSOD1 was determined by nitroblue tetrazolium (NBT) reaction [63] in different combinations. The 3 mL reaction mixture contained 39 mM L-methionine 1.5 mL, 225 μM nitroblue tetrazolium (NBT) 0.3 mL, 8 μM riboflavin (dissolve in 30 μM EDTA-Na$_2$ buffer) 0.3 mL, 10 μL purified enzyme and 50 mM potassium phosphate buffer (pH 7.8) 890 μL. The reaction was initiated by illuminating the reaction mixture for 20 min, and photochemically produced superoxide reacted with NBT. Absorbance of formazan, the product of NBT reduction, was then recorded at 560 nm. One unit of SOD activity was defined as the amount of enzyme that caused 50% of the maximum inhibition of NBT reduction. These experiments were repeated three times. A standard curve of protein concentration was obtained with bovine serum albumin as standard [64].

*In vivo*, we determined the activity of CuZnSOD by using the CuZnSOD assay kit (colorimetry) (Jian Cheng, Nanjing, China, A001-4-1) according to the protocol of manufacturer. Weigh 0.2 g plant tissue sample accurately, add 4 times volume homogenate medium according to mass (g)-volume(ml) ratio of 1:4, cut tissue to small pieces, make homogenate in ice-water bath. Centrifugate at 3500 rpm for 10 min, take supernatant for assay. Take 0.1 ml 20% homogenate supernatant, add 0.2 ml homogenate medium (equals to 3 times dilution), mix sufficiently, take 3 samples of different volumes (10 μl, 30 μl, 50 μl), do pre-test according to operation table in order to determine optimal sample volume. Curve appears direct proportion while inhibition percentage is between 15–55%. Take the tube which inhibition percentage is between 45% to 50% as optimal sample volume. Use xanthine and xanthine oxidase reaction system to produce superoxide anion radicals ($O_2^-$), the latter will oxidate hydroxylamine to form nitrite, appears prunosus color under effect of chromogenic agent, its absorbance can be measured by visible range spectrophotometer. If sample to assay contains SOD, then it has a narrow spectrum depressant effect for superoxide anion radicals, as result, absorbance in sample tube will be lower than absorbance in contrast tube, SOD activity can be calculated by formula. MnSOD and FeSOD loss activity in pretreated samples and CuZnSOD activity keeps stable. These experiments were repeated three times.

## Barley stripe mosaic virus (BSMV)-mediated silencing

Based on the cloned *PstGSRE4* and *TaCZSOD2* genes, non-conserved regions were analyzed, and Premier Primer 5.0 was used to design gene silencing vector primers. According to

previously described methods [65], two fragments of *PstGSRE4* or *TaCZSOD2* were cloned and inserted into BSMV to produce BSMV:PstGSRE4-1/2as, BSMV:TaCZSOD2-1/2as. The wheat phytoene desaturase gene (PDS) was silenced as a positive control. BSMV:α and BSMV:β were mixed with BSMV:γ or recombinant γ-gene, in 1:1:1, and then the appropriate amount of FES buffer (2.613g dipotassium phosphate, 1.877 g glycine, 0.5 g sodium pyrophosphate, 0.5 g diatomite, 0.5 g porphyritic soil, 50 ml constant volume, 20 min sterilization by autoclaving) was added. Each independent experiment set FES buffer as a negative control, BSMV:γ as a blank control and BSMV:γ-TaPDS as positive controls for about 10 d to observe the symptoms of virus infection. After 10 to 14 d following inoculation, *Pst* races CYR23 and CYR31 (fresh urediniospores were collected from the infected leaves of Su11 that were grown at 16˚C in artificial climate chamber) were separately inoculated on the fourth leaf of wheat plants, which were placed in a dark and high humidity environment at 12˚C for 24 h, then grown in a normal 16/8 h light-dark cycle. The fourth leaves were sampled at 24, 48 or 120 hpi for assessment of silencing efficiency and histological observation. The phenotypes of the fourth leaves were photographed at 12 d after inoculation with *Pst*. These experiments were repeated three times.

## Cytological observations of fungal growth and host response

The observation of necrotic death area hyphae and $H_2O_2$ detection assay were performed as previously described [26]. Leaf segments were fixed and decolorized in a mixture of acetic acid/ethanol (1:1) for 3 d. Autofluorescence of mesophyll cells was observed to determine necrotic death area using epifluorescence microscopy (excitation filter, 485 nm; dichromic mirror, 510 nm; barrier filter, 520 nm). $H_2O_2$ accumulation was detected by staining with DAB (Amresco, Solon, OH, USA). Hyphae were stained with WGA conjugated to Alexa-488 (Invitrogen, Carlsbad, CA, USA) and observed under blue-light excitation (excitation wavelength 450–480 nm, emission wavelength 515 nm). Only the site where an appressorium had formed over a stoma was considered to be a successful penetration. The $H_2O_2$ accumulation, necrotic areas, hyphal length, and hyphal areas were observed with a BX-53 microscope (Olympus) and calculated using DP-BSW software.

## Western blotting

Proteins were separated by SDS-PAGE. Gels were blotted onto a PVDF membrane (Merck Millipore, Burlington, MA, USA) with transfer buffer at 64V for 2h. Membranes were blocked for 1 h at room temperature, followed by washing. The antibodies—anti-GFP (1:2,000; #A02020; Abbkine, Wuhan, China), anti-RFP (1:2,000; #A02120; Abbkine), anti-His (1:3,000; #A02050; Abbkine), or anti-GST (1:2,000; #A02030; Abbkine)—were added and incubated at 4˚C overnight, followed by three washes. Membranes were then incubated with goat anti-mouse antibody (ab6789; Abcam), or goat anti-rabbit (ab205718; Abcam) at a ratio of 1:10,000 in the blotting buffer at room temperature for 2 h. After three washes, membranes were incubated with chemiluminescence HRP substrate (#WBKLS0100, Merck Millipore) for 5 min, and then visualized by excitation at 780 or 800 nm.

## Determination of the accumulation of $O_2^-$ and $H_2O_2$

*In vivo*, we determined the content of $O_2^-$ by using the $O_2^-$ assay kit (SA-2-G, Comin Biotechnology, Suzhou, China) according to the protocol of manufacturer. Weigh 0.1 g inoculated leaves in different hours accurately, add 10 times 65 mM phosphate buffer (pH 7.8) according to mass (g)- volume(mL) ratio of 1:10, cut tissue to small pieces, make homogenate in ice-water bath. Centrifugate at 10000 g for 20 min, take supernatant for assay. The 900 μL reaction mixture contained 0.5 mL homogenate, 0.4 mL 10 mM hydroxylamine solution, 37˚C for 20

min. Then add 0.3 mL 17mM 4-aminobenzenesulfonic acid and 0.3 mL 7mM α-naphthyl-amine, 37˚C for 20 min. Then add 0.5 mL 1 chloroform, centrifugate at 8000 g for 5 min, take 1 mL supernatant for assay. Recorded absorbance at 530 nm against a distilled water blank.

We determined the content of $H_2O_2$ by using the $H_2O_2$ assay kit (A064-1-1, Comin Biotechnology, Suzhou, China) according to the protocol of manufacturer. Weigh 0.1 g inoculated leaves in different hours accurately, add 10 times propanone according to mass (g)—volume (mL) ratio of 1:10, cut tissue to small pieces, make homogenate in ice-water bath. Centrifugate at 8000 g for 10 min, take supernatant for assay. The 1.3 mL reaction mixture contained 1 mL homogenate, 0.1 mL titanic sulfate solution and 0.2 mL ammonium water, centrifugate at 8000 g for 5 min, take sediment for assay. Then add 1 mL sulphuric acid solution to dissolve the sediment, let stand for 10 min at the room temperature. Recorded absorbance at 415 nm against a distilled water blank. These experiments were repeated three times.

## Oxidative burst measurement

Leaves from 6-wk-old WT and *TaCZSOD2*-knockdown or *TaCZSOD2*-overexpression transgenic lines were sliced into 10 mm$^2$ discs, and maintained overnight in water in a 96-well plate. Then, the leaf discs were treated with 200 μL of solution containing 8nM chitin (hexa-N-acetyl-chitohexaose), 20 μg/ml peroxidase (Sigma-Aldrich) and 20 nM luminol. Luminescence was recorded for 30 min using a multiscan spectrum. Each data point consisted of six replicates. These experiments were repeated three times.

## Phylogenetic relationship analysis

Multiple alignment was performed by Muscle in MEGA6.0 [66]. The phylogenetic relationship was inferred based on the multiple alignment in MEGA6.0 by the Maximum Likelihood (ML) method based on LG model with bootstrap 1000. The unrooted tree was performed by Interactive Tree of Life (IToL) Version 3.2.3 (http://itol.embl.de/).

## Statistical analyses

Statistical analyses of each treatment were performed with the statistical software version package of IBM SPSS Statistics 21 (IBM SPSS Statistics, IBM Corporation, Armonk, New York, USA).

## Supporting information

**S1 Fig. PstGSRE4 is a Glycine- and Serine-rich secreted protein. (A)** Sequence analysis indicates that PstGSRE4 is a glycine- and serine-rich secreted protein. Triangles indicate serine and asterisks represent glycine residues. Multi-sequence alignment of PstGSRE4 and other three glycine- and serine-rich secreted proteins was performed using CLC Sequence Viewer. **(B)** The motif of PstGSRE4 (PSTCY32_07414) and other three glycine- and serine-rich secreted proteins were predicted by MEME suit (http://meme-suite.org/). The black borders represent m9 region of PstGSRE1 (PSTCY32_24327). **(C)** PstGSRE4 cannot interact with TaLOL2. Only the yeast co-expressing PstGSRE4 and TaCZSOD2 or PstGSRE1 and TaLOL2 grew on the medium SD-Trp-Leu-His-Ade and yielded X-α-gal activity. Yeast strains co-expressing PstGSRE4 and TaLOL2 or PstGSRE1 and TaCZSOD2 cannot grow on the medium SD-Trp-Leu-His-Ade. This experiment was repeated three times. (TIF)

**S2 Fig. Relative transcript levels of *PstGSRE4* at different *Pst* infection stages.** Wheat leaves (Suwon11) inoculated with freshly collected urediniospores (CYR32) were sampled at different

time points according to the infection stage of *Pst*. US (Urediniospores) was used as a control. Relative transcript levels of *PstGSRE4* were calculated by the comparative threshold ($2^{-\Delta\Delta CT}$) method. The quantitative RT-PCR values were normalized to the expression level for *PstEF-1*. The transcript level of *PstGSRE4* at US stage was standardized as 1. Values represent the means ± SE of three independent replicates. Differences between time-course points were assessed using Student's *t*-test. Double asterisks indicate $P < 0.01$.
(TIF)

**S3 Fig. PstGSRE4 has functional signal peptide.** Functional validation of the putative N-terminal signal peptide of PstGSRE4 using the yeast invertase secretion assay. Yeast YTK12 strains carrying pSUC2-SP (*Avr1b*) and pSUC2-SP (*PstGSRE4*), which express two different signal peptides fused in frame to the mature invertase gene SUC2, were able to grow in YPRAA (Yeast-Peptone-Raffinose-Antimycin A) medium with raffinose as sole carbon source. YTK12 or YTK12 strains carrying empty vector pSUC2T7M13ORI were used as negative control. Invertase activity was detected with 2,3,5-triphenyltetrazolium chloride (TTC). The red color indicates invertase activity.
(TIF)

**S4 Fig. PstGSRE4 inhibits Pst322- and Bax-induced cell death. (A)** PstGSRE4 suppressed Pst322-induced cell death. Photos of *N. benthamiana* leaves were taken under ultraviolet light. **(B)** PstGSRE4 suppressed Bax-induced cell death. Photos of *N. benthamiana* leaves were taken under ultraviolet light. **(C)** Western blot with anti-HA antibody was performed to show normal expression of eGFP-HA (25kDa), Pst18363-HA (20kDa), PstGSRE4-HA (22kDa), Pst322-HA (20kDa) and Bax-HA (25kDa) in tobacco leaves.
(TIF)

**S5 Fig. PstGSRE4 suppresses Pst322- and Bax-triggered cell death by decreasing ROS accumulation. (A)** $H_2O_2$ production in *N. benthamiana* leaves was determined by DAB staining. The measurement was performed at 3 d after infiltration with Pst322 or Bax (left). And overexpression of Bax or Pst322 in *Nicotiana benthamiana* triggered programmed cell death (PCD) at 4 d after infiltration with Pst322 or Bax (right). **(B)** Content of $H_2O_2$ in *N. benthamiana* leaves was determined at 3 d after infiltration with Pst322 or Bax. Values represent the means ± SE of three independent samples. **(C)** $H_2O_2$ production in *N. benthamiana* leaves was determined by DAB staining. *N. benthamiana* leaves were infiltrated with *A. tumefaciens* cells carrying the PstGSRE4-HA or $MgCl_2$ buffer, followed after 24 h by infiltration with *A. tumefaciens* cells carrying the Bax or Pst322. The measurement was performed at 3 d after infiltration with Bax or Pst322 (left). Overexpression of PstGSRE4 in *Nicotiana benthamiana* suppressed programmed cell death (PCD) triggered by Bax or Pst322 at 4 d (right). **(D)** Content of $H_2O_2$ in *N. benthamiana* leaves was determined at 3 d after infiltration with PstGSRE4, $MgCl_2$, PstGSRE4/Bax, PstGSRE4/Pst322, $MgCl_2$/Bax or $MgCl_2$/Pst322. Values represent the means ± SE of three independent samples.
(TIF)

**S6 Fig. Overexpression of *PstGSRE4* in wheat suppresses PTI-associated callose deposition and $H_2O_2$ accumulation. (A)** Phenotypes of wheat leaves (Suwon11) inoculated with CYR23 after being injected with *Pseudomonas fluorescens* strain EtHAn alone or carrying plasmids pEDV6-RFP (a red autofluorescent protein, DsRed), or pEDV6-PstGSRE4 at 14 dpi. HR, hypersensitive response. **(B-C)** Wheat leaves inoculated as above were examined for callose deposition by epifluorescence microscopy after aniline blue staining. Scale bars, 100 μm. The average number of callose deposits per $mm^2$ at 24 and 48 hpi was counted using ImageJ software. Values represent the means ± SE (n = 20). **(D-E)** $H_2O_2$ production in leaves infiltrated

with EtHAn, EtHAn pEDV6-RFP, or EtHAn pEDV6-PstGSRE4 at 24 and 48 hpi with *Pst*. SV, substomatal vesicle. Tissues were stained with DAB. Scale bars, 20 μm. The amount of $H_2O_2$ production was measured by calculating the DAB-stained area at each infection site using DP-BSW software. Values represent the means ± SE (n = 30). Asterisks indicate a significant difference ($P < 0.05$) relative to the control sample according to Student's *t*-test, double asterisks indicate $P < 0.01$.
(TIF)

**S7 Fig. BSMV-mediated host-induced gene silencing (HIGS) of *PstGSRE4* reduces virulence of *Pst*. (A)** Two specific sequence regions were selected for BSMV-mediated transient silencing. **(B)** Wheat leaves infected with BSMV:TaPDS, which showed photobleaching phenotype, was used as control. Mild chlorotic mosaic symptoms were observed on the fourth leaves of the wheat inoculated with BSMV:γ, BSMV:PstGSRE4-1as, and BSMV:PstGSRE4-2as. Phenotypes of the fourth leaves of knockdown plants or control plants inoculated with *Pst* race CYR32 at 12 dpi. **(C)** Relative transcript levels of *PstGSRE4* in *PstGSRE4*-knockdown plants challenged by CYR32. *PstEF-1* was used for normalization. Values represent the means ± SE (n = 3). **(D)** Ratio of fungal to wheat nuclear content using fungal *PstEF-1* and wheat *TaEF-1α* genes, respectively. Genomic DNA was extracted from the second leaf from three different plants at 14 dpi. Values represent the means ± SE (n = 3). Differences were assessed using Student's *t*-test, and asterisks indicate $P < 0.05$, double asterisks indicate $P < 0.01$.
(TIF)

**S8 Fig. Histological changes of *Pst* growth in *PstGSRE4*-knockdown plants. (A)** Fungal growth at 24, 48 and 120 hpi in wheat leaves inoculated with BSMV:γ, BSMV:PstGSRE4-1as and BSMV:PstGSRE4-2as. **(B-C)** Hyphal lengths (48 and 120 hpi) and colony sizes (120 hpi) in *PstGSRE4*-knockdown plants were stained with WGA and quantified with DP-BSW software. SV, substomatal vesicle. HMC, haustorial mother cell. IH, infection hypha. Values represent the means ± SE (n = 30, n = 20). Differences were assessed using Student's *t*-test, and asterisks indicate $P < 0.05$.
(TIF)

**S9 Fig. Cytological observation of host response in *PstGSRE4*-RNAi wheat leaves inoculated with *Pst* race CYR31. (A)** Diagram showing the RNAi cassette in the wheat transformation construct pAHC25-*PstGSRE4*-RNAi. Ubi1, maize ubiquitin1 promoter. Adh1, *Zea mays* alcohol dehydrogenase 1. Noc Term, Nos terminator. Bar, Biolaphos resistance gene. LB, left border. RB, right border. **(B)** Transgenic plants were analyzed by genomic PCR for the presence of the selectable marker *Bar* gene and the fragment of the RNAi cassette (*PstGSRE4*). **(C)** Leaves inoculated with *Pst* race CYR31 were sampled at 48 and 120 hpi and examined under epifluorescence after staining with WGA conjugated to Alexa-488. SV, substomatal vesicle. HMC, haustorial mother cell. IH, infection hypha. Scale bars, 20 μm (left) and 50 μm (right). **(D)** $H_2O_2$ accumulation was measured in transgenic plants at 72 and 120 hpi. DAB was used to detect $H_2O_2$ viewed under differential interference contrast optics. Scale bars, 20 μm. **(E)** Content of $O_2^-$ accumulation in different transgene lines at 6, 12, 24 and 48 hpi. Values represent the means ± SE of three independent samples. **(F)** Quantification of $H_2O_2$ accumulation in different transgenic lines at 72 and 120 hpi. Values represent the means ± SE (n = 30). Differences were assessed using Student's *t*-test. Asterisk indicates $P < 0.05$.
(TIF)

**S10 Fig. Molecular analysis of *PstGSRE4*-overexpression wheat. (A)** Diagram showing the overexpression cassette in the wheat transformation construct pCAMBIA3301-*PstGSRE4*-overexpression. LB, left border. RB, right border. **(B-C)** Transgenic plants were analyzed by

genomic PCR for the presence of the selectable marker *Bar* gene and the pCAMBIA3301 primer. **(D)** Expression of *PstGSRE4* in T$_3$ lines (L2 and L3) was analyzed by RT-PCR. Expression of *TaEF-1α* showed equal loading. **(E)** Content of O$_2^-$ in different transgenic lines at 6, 12, 24 and 48 hpi. Values represent the means ± SE of three independent samples. **(F)** Quantification of H$_2$O$_2$ accumulation in different transgenic lines at 48 hpi. **(G)** H$_2$O$_2$ accumulation at infection sites were observed by microscopy after DAB staining. SV, substomatal vesicle. Scale bars, 20 μm. Values represent the means ± SE (n = 20). **(H)** Observation of necrotic cell death by epifluorescence in transgenic plants. NC, necrotic cell death. SV, substomatal vesicle. Scale bars, 20 μm. Differences were assessed using Student's *t*-test, and asterisks indicate $P < 0.05$, double asterisks indicate $P < 0.01$.
(TIF)

**S11 Fig. Phylogenetic analysis of SOD in *Arabidopsis thaliana*, rice and wheat.** Branches are labeled with protein names and GenBank accession number. Os, *Oryza sativa*. At, *Arabidopsis thaliana*. The red line represents TaCZSOD2-7A. The tree was created with bootstrap of 1000 by maximum likelihood method in MEGA6. And the tree was drawn using Interactive Tree of Life (IToL).
(TIF)

**S12 Fig. Sequence analysis of *TaCZSOD2*.** Multi-alignment of the coding sequences of the three copies of *TaCZSOD2* in the genome database of wheat cultivar Chinese Spring.
(TIF)

**S13 Fig. PstGSRE4 interacts with TaCZSOD2 in the cytoplasm of *N. benthamiana*. (A)** Confocal microscopy images showing the subcellular localization of PstGSRE4 and TaCZSOD2. Western blotting analysis shows the total protein of PstGSRE4-GFP and TaCZSOD2-GFP in *N. benthamiana* leaves. Scale bars, 50 μm. **(B)** Co-localization of PstGSRE4-RFP (48 kDa) and TaCZSOD2-GFP (47 kDa) in *N. benthamiana*. In all panels, proteins were expressed in *N. benthamiana* through agroinfiltration. Fluorescence was detected in epidermal cells of the infiltrated leaves by confocal microscopy at 48 h after agroinfiltration (prior to any cell death). Scale bars, 50 μm. Localization of RFP-fusion of PstGSRE4 was uniformly located in the cytoplasm and nucleus, GFP-fusion of TaCZSOD2 was primarily located in the chloroplast, but also in the cytoplasm. When expressed together, co-localization of PstGSRE4 and TaCZSOD2 accumulated in the cytoplasm. Western blotting analysis shows the total protein of PstGSRE4-RFP and TaCZSOD2-GFP in *N. benthamiana* leaves. **(C)** Co-localization of GFP fusion of chloroplast transit peptide-deleted TaCZSOD2 (ΔTP-TaCZSOD2) and RFP-fusion of PstGSRE4 in *N. benthamiana*. They accumulated in the cytoplasm and nucleus. Scale bars, 50 μm. **(D)** Co-expressed GFP fusion of ΔTP-TaCZSOD2 with RFP-fusion of PstGSRE4, GFP fusion of TaCZSOD2 with RFP-fusion of PstGSRE4, GFP fusion of ΔTP-TaCZSOD2 with RFP, GFP fusion of TaCZSOD2 with RFP in *N. benthamiana*, and detected the activity of CuZnSOD. RFP was expressed as control. Values represent the means ± SE of three independent samples. These experiments were repeated three times and obtained the similar result.
(TIF)

**S14 Fig. *TaCZSOD2* positively regulates wheat resistance against *Pst* in an ROS-dependent manner. (A)** Two specific sequence regions were selected for BSMV-mediated transient silencing. **(B)** Wheat leaves (Suwon11) inoculated with freshly collected urediniospores (CYR23 and CYR31) were sampled at different time points according to the infection stage of *Pst*. Relative transcript levels of *TaCZSOD2* were calculated by the comparative threshold ($2^{-\Delta\Delta CT}$) method. The quantitative qRT-PCR values were normalized to the expression level for *TaEF-1α*. Differences between time-course points were assessed using Student's *t*-tests.

Asterisks indicate $P < 0.05$, double asterisks indicate $P < 0.01$. Values represent the means ± SE (n = 3). **(C)** After inoculated with CYR31, ratio of fungal to wheat nuclear content using fungal *PstEF-1* and wheat *TaEF-1α* genes, respectively. Values represent the means ± SE (n = 3). **(D)** Content of $O_2^-$ accumulation in *TaCZSOD2*-knockdown plants at 6, 12, 24 and 48 hpi. Values represent the means ± SE of three independent samples. **(E)** Quantification of $H_2O_2$ accumulation in *TaCZSOD2*-knockdown plants at 48 hpi. Values represent the means ± SE (n = 20). **(F)** $H_2O_2$ accumulation at infection sites was observed by microscopy after DAB staining. SV, substomatal vesicle. Scale bars, 10 μm. **(G)** Observation of necrotic cell death by epifluorescence in *TaCZSOD2*-knockdown wheat plants. NC, necrotic cell death. SV, substomatal vesicle. Scale bars, 20 μm.
(TIF)

**S15 Fig. Transient overexpression of *TaCZSOD2* increases wheat resistance against *Pst* in an ROS-dependent manner. (A)** Phenotypes of pEDV6-RFP- and pEDV6-TaCZSOD2--treated wheat plants inoculated with the virulent *Pst* race CYR31 at 12 dpi. **(B)** Quantification of $H_2O_2$ accumulation at 48 hpi in pEDV6-RFP- and pEDV6-TaCZSOD2-treated wheat plants inoculated with virulent *Pst* race CYR31. Values represent the means ± SE (n = 30). **(C)** Quantification of necrotic cell death area in *TaCZSOD2*-overexpression plants at 48 hpi. Values represent the means ± SE (n = 20). Differences between time-course points were assessed using Student's *t*-tests. Asterisks indicate $P < 0.05$, double asterisks indicate $P < 0.01$. **(D)** $H_2O_2$ accumulation was observed by microscopy after DAB staining and necrotic cell death observed by epifluorescence. SV, substomatal vesicle. NC, necrotic cell death. Scale bars, 20 μm.
(TIF)

**S16 Fig. Molecular analysis of *TaCZSOD2*-overexpression transgenic wheat. (A)** Diagram showing the overexpression cassette in the wheat transformation construct CUB-*TaCZSOD2*-overexpression. LB, left border. RB, right border. **(B-C)** Transgenic plants were analyzed by genomic PCR and western blotting. **(D)** Relative transcript levels of *TaCZSOD2* in *TaCZSOD2*-overexpression plants challenged by CYR31. *TaEF-1α* was used for normalization. Values represent the means ± SE (n = 3). **(E)** Content of $O_2^-$ accumulation in different transgene lines at 6, 12, 24 and 48 hpi. Values represent the means ± SE of three independent samples. **(F)** $H_2O_2$ accumulation at infection sites were observed by microscopy after DAB staining. SV, substomatal vesicle. Scale bars, 20 μm. **(G)** Observation of necrotic cell death by epifluorescence in transgenic plants. NC, necrotic cell death. SV, substomatal vesicle. Scale bars, 20 μm. **(H)** Quantification of $H_2O_2$ accumulation in different transgenic lines at 48 hpi. Values represent the means ± SE (n = 30). Differences between time-course points were assessed using Student's *t*-tests. Asterisks indicate $P < 0.05$.
(TIF)

**S17 Fig. TaCZSOD2 increases ROS accumulation. (A)** Relative transcript levels of *TaCZSOD2* in *TaCZSOD2*-knockdown plants were calculated by the comparative threshold ($2^{-\Delta\Delta CT}$) method. The quantitative qRT-PCR values were normalized to the expression level for *TaEF-1α*. Values represent the means ± SE (n = 3). **(B)** Reactive oxygen species (ROS) burst induced by 8 nM chitin in discs of *TaCZSOD2*-knockdown and WT leaves. Values represent the means ± SE (n = 6). **(C)** Relative transcript levels of *TaCZSOD2* in *TaCZSOD2*-overexpression transgene lines were calculated by the comparative threshold ($2^{-\Delta\Delta CT}$) method. The quantitative qRT-PCR values were normalized to the expression level for *TaEF-1α*. Values represent the means ± SE (n = 3). **(D)** Reactive oxygen species (ROS) burst induced by 8 nM chitin in discs of *TaCZSOD2*-overexpression and WT leaves. Values represent the means ± SE

(n = 6).
(TIF)

**S1 Table. BlastP analysis of protein sequence similarities between PstGSRE4 and homologous proteins.**
(XLSX)

**S2 Table. PstGSRE4 is a specific effector in rust fungi.**
(XLSX)

**S3 Table. Partial candidate targets of PstGSRE4 through Y2H screening.**
(XLSX)

**S4 Table. Primers used in this study.**
(XLSX)

**S1 File. Alignment of the proteins for phylogenetic analysis of SOD.**
(DOCX)

**S1 Data. Statistics to support this study.**
(XLS)

## Acknowledgments

We thank Professor Larry Dunkle for editing the manuscript and Professor Daolong Dou for helpful suggestions. We thank Qiong Zhang, Fengping Yuan, Hua Zhao and Xiaona Zhou of State Key Laboratory of Crop Stress Biology for Arid Areas for their technical support.

## Author Contributions

**Conceptualization:** Cong Liu, Zhensheng Kang, Jun Guo.

**Data curation:** Cong Liu, Yunqian Wang.

**Formal analysis:** Cong Liu, Yunqian Wang, Yanfeng Wang, Yuanyuan Du, Chao Song, Ping Song, Xingxuan Bai, Lili Huang, Jia Guo, Jun Guo.

**Funding acquisition:** Zhensheng Kang, Jun Guo.

**Investigation:** Cong Liu, Jun Guo.

**Methodology:** Cong Liu, Qian Yang, Fuxin He, Jun Guo.

**Project administration:** Cong Liu, Yunqian Wang, Yanfeng Wang.

**Resources:** Cong Liu, Yunqian Wang, Yanfeng Wang.

**Software:** Cong Liu, Yanfeng Wang, Jia Guo.

**Supervision:** Jun Guo.

**Validation:** Cong Liu, Yunqian Wang, Yanfeng Wang, Yuanyuan Du, Chao Song, Jun Guo.

**Visualization:** Cong Liu.

**Writing – original draft:** Cong Liu.

**Writing – review & editing:** Cong Liu, Qian Yang, Jia Guo, Jun Guo.

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
