## [Decision Letter · Decision Letter 0]

14 Mar 2022

Dear Prof. Guo,

Thank you very much for submitting your manuscript "Glycine-serine-rich effector PstGSRE4 in Puccinia striiformis f. sp. tritici inhibits the activity of copper zinc superoxide dismutase to modulate immunity in wheat" for consideration at PLOS Pathogens. As with all papers reviewed by the journal, your manuscript was reviewed by members of the editorial board and by several independent reviewers. A number of additional experiments are requested to address the concerns raised by reviewers. Detailed information for methodology and number of replicates in many of the experiments is missing. In light of the reviews (below this email), we would like to invite the resubmission of a significantly-revised version that takes into account the reviewers' comments.

We cannot make any decision about publication until we have seen the revised manuscript and your response to the reviewers' comments. Your revised manuscript is also likely to be sent to reviewers for further evaluation.

Sincerely,

Jie Zhang

Guest Editor

PLOS Pathogens

Hui-Shan Guo

Section Editor

PLOS Pathogens

Kasturi Haldar

Editor-in-Chief

PLOS Pathogens

orcid.org/0000-0001-5065-158X

Michael Malim

Editor-in-Chief

PLOS Pathogens

orcid.org/0000-0002-7699-2064

Reviewer's Responses to Questions

**Part I - Summary**

Reviewer #1: The work presented by Liu et al shows that a new glycine-serine-rich effector, PstGSRE4, reduces H2O2 accumulation and HR areas to facilitate Puccinia striiforms f. sp. Tritici (Pst) infection, and is highly induced during the early stages of infection. Moreover, PstGSRE4 inhibits the enzyme activity of wheat copper zinc superoxide dismutase TaCZSOD2, which acts as a positive regulator of wheat resistance to Pst. Such systematic research results provide new insights into the molecular mechanisms of GSREs of rust fungi in regulating plant immunity. Overall, I think this research is a major advancement in the plant-microbe interaction field, especially the pathogenic mechanism of biotrophic fungi.

Reviewer #2: The paper describes the function of a glycine-serine-rich effector protein from Puccinia striiformis f.sp. tritici (Pst). The authors demonstrated that this protein inhibits the activity of one of the wheat copper zinc superoxide dismutases (TaCZSOD2) and this way prevents H2O2 accumulation and promotes virulence. The results are well described and the conclusions taken are correct. The results provide a mechanistic understanding on how fungal pathogens manipulate plant immunity. Although this has been addressed in many works, it still remains to be unveiled. My main concern is related to the information provided in the materials and methods. Sometimes I found this information scarce or not clear enough.

Reviewer #3: Liu et al. present work on the molecular characterization of the effector PstGSRE4 and its host target TaCuZnSOD2. Based on previous work on the effector PstGSRE1, they identified members of the glycine/serine-rich effector family (GSRE). PstGSRE4 was found to lack the m9 domain previously found in PstGSRE1 and yeast two-hybrid experiments found that PstGSRE4 did not interact with TaLOL2, the target of PstGSRE1. Secretion assays in yeast found that the signal peptide of PstGSRE4 is functional. PstGSRE4 is expressed early in the interaction with wheat and most highly expressed between 24-48 hpi. They found that infiltration of Pseudomonas fluorescence carrying PstGSRE4 reduced callose formation, indicating that PstGSRE4 can contribute to the suppression of PTI responses. Similar results were observed for transient expression in wheat using Agrobacterium-mediated transformation. Host induced gene silencing of PstGSRE4 using two distinct fragments were integrated into BSMV and caused a reduction in PstGSRE4 transcript accumulation and led to reduced Pst biomass. Transgenic wheat lines expressing an RNAi construct targeting PstGSRE4. Similar to HIGS experiments with BSMV, the stable RNAi caused a reduction in Pst development and an increase in ROS formation (H2O2). Transgenic wheat lines carrying overexpression constructs of PstGSRE4 led to a decrease in ROS formation and increase in Pst development. A yeast two-hybrid screen identified TaCuZnSOD2 as a putative interactor. The interaction was confirmed with expression in E. coli based on pulldown of recombinant PstGSRE4 and TaCuZnSOD2 and Co-IP in Nicotiana benthamiana. Fluorescently tagged PstGSRE4 and TaCuZnSOD2 were found to colocalise in the cytoplasm. Transient silencing of TaCuZnSOD2 using BSMV found that it positively regulates immunity to Pst to an avirulent isolate (CYR23) but not a virulent isolate (CYR31). Enzymatic activity was also reduced. Expression of recombinant proteins of PstGSRE4 and TaCuZnSOD2 showed that PstGSRE4 inhibits the activity of TaCuZnSOD2 in vitro and in vivo. Collectively, this is a systematic study that used almost every available molecular technique to investigate the molecular function of PstGSRE4 and its target, TaCuZnSOD2.

**Part II – Major Issues: Key Experiments Required for Acceptance**

Reviewer #1: (No Response)

Reviewer #2: - In the introduction, it will be important to include information about the differences in activity between O2- and H2O2 since this is critical to understand the results of the work.

- My main concern is related to the description on how the activity oassays of CuZnSOD were performed. Was the protein purified? If so, the methodology should be described. How was the NBT reaction performed? How was the concentration of the proteins determined?

- In the infection assays, line 639: How was the inoculum obtained?

- Determination of accumulation of O2 and H2O2. How was it performed? what is the kit consisting in?

- For most of the experiments, what is considered a biological replicate? Were the experiments independent?

- I could not find information about how the qPCR was performed.

- Why the estimation of H2O2 content was performed using different methodologies in fig1 and 2? Both experiments should use the same methodology.

- The legend of figure 6 is not complete. Specially 6D and 6F need more information about the experiment.

Reviewer #3: No single major issue was identified, but considerable minor issues were found that collectively become a major issue, particularly in data presentation.

**Part III – Minor Issues: Editorial and Data Presentation Modifications**

Reviewer #1: Here I list several points or suggestions that I believe would improve the manuscript.

1. In the manuscript, the author described that PstGSRE4 inhibits the enzyme activity of TaCZSOD2 to facilitate Pst infection, but the part of “TaCZSOD2 positively regulates wheat resistance against Pst”, showed that transcript levels of TaCZSOD2 were up-regulated at 12, 48, and 96 hpi with the avirulent Pst race CYR23, and up-regulated at 96 and 120 hpi with the virulent Pst race CYR31. Here, I have several questions. First, does the transcript levels increasing represent the increasing the protein level? Second, CYR23 is an avirulent strain, while CYR31 is a virulent strain, the authors need to illustrate why these two strains cause different expression profiles of TaCZSOD2. Third, which strain includes the effector PstGSRE4? In the model, I can see PstGSRE4 is from the CYR31 strain, but the part of “Relative transcript levels of PstGSRE4 at different Pst infection stages”, showed that PstGSRE4 is from CYR32.

2. The author should add some experiments to prove that PstGSRE4 depends on which pathway to suppresses Pst322- and Bax-induced cell death.

3. ROS burst as an important signal after pathogen infection, and the manuscript proved that TaCZSOD2 is a positive regulator of wheat immunity. I would suggest to identify the ROS burst level in the knockdown and overexpression lines of TaCZSOD2 treated with flg22 or Chitin.

4. Whether GSRE1 interact with CZSOD2 in the Y2H system?

5. Please correct the spelling mistakes in lines 178, 518, 586, 653, et al.

6. Lines 254 and 255, “The hyphal areas during infection of L19 and L76 were significantly reduced at 48 and…..”, I think “significantly reduced” is not an acceptable description based on the result showed in Fig. 1D.

7. The description is not clear in lines 445-447.

8. “d” or “day”, for example in line 564 and line 571, they should be uniform.

Reviewer #2: - A large section of the abstract contains information about a previosly identified effector protein. I would recommend to reduce this part to a minimum.

- Line 325: Indicate what is the meaning of TP

- Material and methods. There are several references of the products that are missing : Line 609, Line 613

- Lines 490-493. What do the authors mean with "at certain levels"? Could this sentence be more precise?

- In figure 1D, hyphal areas are quantified. What is this exactly and how was it performed?

Reviewer #3: For all bar plots in the manuscript, raw data points should be included within each figure. This is even more important when only three biological replicates were performed. Also include the number of fields that were analysed for all the microscopy experiments.

L 377. Change transient for stable transformation.

Figure 3A. Include a negative control for TaCZSOD2 in the yeast-two-hybrid experiment.

Figure 3B. There doesn’t seem to be enrichment of PstGSRE4-GST in the GST pulldown experiment with TaCZSOD1-His. Do you have any idea of why this might have happened? The pulldown should be enriching the GST-tagged effector.

Figure 3C. PstGSRE4-GFP is not present in the input of the CoIP experiments, yet it is detected in other experiments (e.g. Fig. S4C, Fig. S12A). Can you explain why this might be?

Figure 4C. "BSMV:" should be removed. The expression of the other TaCZSODs was checked, but these were not targeted with BSMV.

S1 Fig. State which of the four proteins in panel A are GSRE1 and GSRE4. State the number of replicate experiments performed for panel C.

S4 Fig and S12 Fig. Within the figure legend, list the predicted protein sizes for all five recombinant proteins shown in panel C.

The legend title for S5 Fig is incorrect, change ‘S’ to 5. Which wheat accession was used in this experiment?

S6 Fig. State what is shown in Panel A. Is this genomic DNA or cDNA?

S7 Fig. How many sites were evaluated for Panel B/C? State in legend.

S8 Fig. B. Marker instead of Maker.

For length of scale bar, state “left” and “right” for 20 and 50 um, respectively.

S10 Fig. Alignment of the proteins needs to publicly deposited (Figshare) or provided as supplemental data. The number of bootstraps performed should be included the legend. The methods for tree construction were not present in the Materials and Methods. State that this is an unrooted tree. State explicitly in the legend that this is based on a protein alignment.

S12 Fig. State how many experiments were performed and the consistency of the observations.

S13 Fig. State what is shown in Panel A (cDNA I assume).

S15 Fig C. Lane for L9 in western blot seems to be cropped from a different photo. Make a clear differentiation of photos of different blots.

S3 Table. The IWGSC identifiers should be included for all putative interactors.

PLOS authors have the option to publish the peer review history of their article (what does this mean?). If published, this will include your full peer review and any attached files.

Reviewer #1: No

Reviewer #2: No

Reviewer #3: **Yes: **Matthew Moscou
---

## [Decision Letter · Decision Letter 1]

11 Jun 2022

Dear Dr. Guo,

Thank you very much for submitting your manuscript "Glycine-serine-rich effector PstGSRE4 in Puccinia striiformis f. sp. tritici inhibits the activity of copper zinc superoxide dismutase to modulate immunity in wheat" for consideration at PLOS Pathogens. As with all papers reviewed by the journal, your manuscript was reviewed by members of the editorial board and by several independent reviewers. The reviewers appreciated the attention to an important topic. Based on the reviews, we are likely to accept this manuscript for publication, providing that you modify the manuscript according to the review recommendations.

Reviewer 4 raised a minor issue on the actitvty of TaCZSOD1/3/4. Please address or discuss about this issue.

Sincerely,

Jie Zhang

Guest Editor

PLOS Pathogens

Hui-Shan Guo

Section Editor

PLOS Pathogens

Kasturi Haldar

Editor-in-Chief

PLOS Pathogens

orcid.org/0000-0001-5065-158X

Michael Malim

Editor-in-Chief

PLOS Pathogens

orcid.org/0000-0002-7699-2064

Reviewer Comments (if any, and for reference):

Reviewer's Responses to Questions

**Part I - Summary**

Reviewer #1: In the revised manuscript, the authors have fully addressed all of my concerns. I believe the data to be convincing and sufficient to demonstrate the role of Puccinia striiformis f. sp. tritici (Pst) secretes Glycine-serine-rich effector protein PstGSRE4 manipulate plant immunity through inhibition of the enzyme activity of wheat copper zinc superoxide dismutase TaCZSOD2, and reduction of H2O2 accumulation and HR areas to facilitate Pst infection. Clearly, many interesting avenues for follow up will emerge from this work from the authors as well as other groups in the field of plant-microbe interaction. I’m looking forward to seeing this work published.

Reviewer #2: The authors replied to all the comments I made on the previous version of the manuscript and I do not have further comments or queestions

Reviewer #4: This study reported a new glycine-serine-rich effector protein lacking the m9-like motif, PstGSRE4, which inhibits the enzyme activity of the wheat TaCZSOD2 to modulate ROS-associated defense responses. By inhibiting the enzyme activity of TaCZSOD2, PstGSRE4 reduces H2O2 accumulation and HR to facilitate Puccinia striiformis f. sp. tritici infection. These findings provide new insights into the molecular mechanisms of GSREs of phytopathogenic fungi in regulating plant immunity.

In this revision, authors have address all concerns and suggestions raised by reviewers. Thus, I think that this revision could be accepted for publication in PLoS Pathogens.

**Part II – Major Issues: Key Experiments Required for Acceptance**

Reviewer #1: NO.

Reviewer #2: (No Response)

Reviewer #4: (No Response)

**Part III – Minor Issues: Editorial and Data Presentation Modifications**

Reviewer #1: NO.

Reviewer #2: (No Response)

Reviewer #4: I have only one minor point that could be discussed. In addition to CZSOD2, the wheat (cultivar Suwon11) contains other three highly conversed TaCZSODs (TaCZSOD1, TaCZSOD3, TaCZSOD4). Since PstGSRE4 can directly interact with TaCZSOD2, and inhibit its’ activity. Why cannot TaCZSOD1/3/4 catalyze the production of H2O2 from O2- during Pst infection？

PLOS authors have the option to publish the peer review history of their article (what does this mean?). If published, this will include your full peer review and any attached files.

Reviewer #1: No

Reviewer #2: No

Reviewer #4: No

Figure Files:

Data Requirements:

Reproducibility:

References:

---

## [Editor Report · Decision Letter 2]

23 Jun 2022

Dear Dr. Guo,

We are pleased to inform you that your manuscript 'Glycine-serine-rich effector PstGSRE4 in Puccinia striiformis f. sp. tritici inhibits the activity of copper zinc superoxide dismutase to modulate immunity in wheat' has been provisionally accepted for publication in PLOS Pathogens.

Best regards,

Jie Zhang

Guest Editor

PLOS Pathogens

Hui-Shan Guo

Section Editor

PLOS Pathogens

Kasturi Haldar

Editor-in-Chief

PLOS Pathogens

orcid.org/0000-0001-5065-158X

Michael Malim

Editor-in-Chief

PLOS Pathogens

orcid.org/0000-0002-7699-2064
---

## [Editor Report · Acceptance letter]

12 Jul 2022

Dear Dr. Guo,

We are delighted to inform you that your manuscript, "Glycine-serine-rich effector PstGSRE4 in Puccinia striiformis f. sp. tritici inhibits the activity of copper zinc superoxide dismutase to modulate immunity in wheat," has been formally accepted for publication in PLOS Pathogens.

Best regards,

Kasturi Haldar

Editor-in-Chief

PLOS Pathogens

orcid.org/0000-0001-5065-158X

Michael Malim

Editor-in-Chief

PLOS Pathogens

orcid.org/0000-0002-7699-2064